

# Critical contribution of chemically diverse carbonyl molecules to the oxidative potential of atmospheric aerosols

Feifei Li[1,2], Shanshan Tang[3], Jitao Lv[1,2], Shiyang Yu[1,2], Xu Sun[1,4], Dong Cao[1], Yawei Wang[1,2,5], Guibin Jiang[1,2,5]

[1]State Key Laboratory of Environmental Chemistry and Eco-toxicology, Research Center for Eco-environmental Sciences, Chinese Academy of Sciences, Beijing 100085, China

[2]College of Resources and Environment, University of Chinese Academy of Sciences, Beijing 100049, China

[3]Collaborative Center for Physics and Chemistry, Institute of International Innovation, Beihang University, Yuhang District, Hangzhou 311115, China

[4]Beijing Urban Ecosystem Research Station, State Key Laboratory of Urban and Regional Ecology, Research Center for Eco-Environmental Sciences, Chinese Academy of Sciences, Beijing 100085, China

[5]School of Environment, Hangzhou Institute for Advanced Study, University of Chinese Academy of Sciences, Hangzhou 310024, China

*Correspondence to*: Jitao Lv (jtlv@rcees.ac.cn) and Yawei Wang (ywwang@rcees.ac.cn)



**Abstract.** Carbonyls have an important effect on atmospheric chemistry and human health because of their high
electrophilicity. Here, high-throughput screening of carbonyl molecules in complex aerosol samples was achieved by
combining targeted derivatization with non-targeted analysis using Fourier transform ion cyclotron resonance mass
spectrometry (FT-ICR MS). Results showed that water-soluble organic matter (WSOM) in $PM_{2.5}$ contains a large variety of
carbonyls (5147 in total), accounting for 17.6% of all identified organic molecules. Compared with non-carbonyl molecules,
carbonyl molecules are more abundant in winter than in summer, and have unique molecular composition and chemical
parameters. For the first time, a significant positive correlation was found between the abundance of carbonyl molecules and
the dithiothreitol (DTT) activities of WSOM, and the elimination of the carbonyl group remarkably reduced the DTT
activities, highlighting the pivotal role of carbonyls in determining the oxidative potential (OP) of organic aerosol. Among
various molecules, oxidized aromatic compounds containing the carbonyl group produced in winter contributed more to the
enhancement of DTT activity, which could be used as potential markers of atmospheric oxidative stress. This study improves
our understanding of the chemical diversity and environmental health effects of atmospheric carbonyls, emphasizing the
need for targeted strategies to mitigate the health risks associated with carbonyl-rich aerosols.

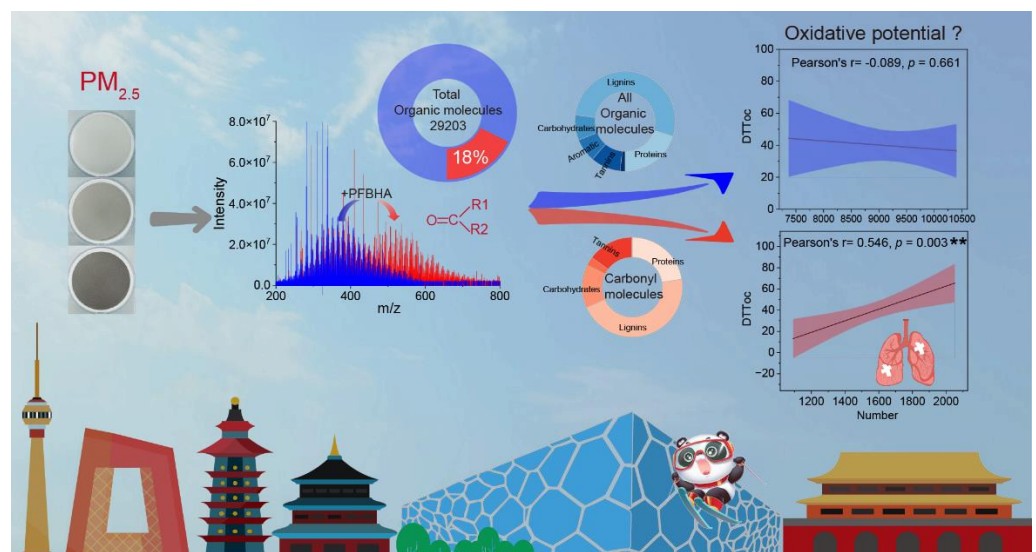

**Graphical abstract**




# 1 Introduction

Atmospheric carbonyls are important components of atmospheric organic compounds, forming not only from primary sources such as vehicle emissions, coal and biomass fuel combustion, and industrial emissions (Liu et al., 2022), but also through secondary reactions between atmospheric oxidants and volatile organic compounds (VOCs) emitted from anthropogenic and biological sources (Claeys et al., 2004; Kalberer et al., 2004a). Meanwhile, carbonyls are also involved in the formation of strong oxidative pollutants such as $O_3$ and peroxyacetyl nitrate (Liu et al., 2022), as well as the formation of secondary organic aerosols (Chen et al., 2023), thus playing a central role in atmospheric chemistry (Lary and Shallcross, 2000). Moreover, carbonyls are reactive and electrophilic compounds, which can react with nucleophilic sites in biomacromolecules, leading to biomacromolecular dysfunction and eventually cytotoxicity (Han et al., 2020; Kumagai et al., 2002; Rappaport et al., 2012).

Due to the key role of carbonyls in atmospheric chemical processes (Ji et al., 2020; Powelson et al., 2014) and its effects on human health (Bora and Shankarrao Adole, 2021; Li et al., 2015), more and more attentions have been paid to atmospheric carbonyls (Chen et al., 2023; Yang et al., 2023; Ye et al., 2023). Currently, high-performance liquid chromatography-ultraviolet/visible detection (HPLC-UV/Vis) (Qian et al., 2019a; Shen et al., 2018; Yu et al., 2023b) or gas chromatography-mass spectrometry (GC-MS) (Ho and Yu, 2004) coupled with carbonyl chemical derivatization are the dominant methods for the analysis of atmospheric carbonyls. These methods mainly focused on atmospheric VOCs such as formaldehyde, acetaldehyde, acetone, keto acids, glyoxal, and methylglyoxal (Liu et al., 2022). However, growing studies suggest that the concentration of carbonyls in the particulate phase was grossly underestimated and plenty of water-soluble and semi-volatile carbonyls with high molecular weight are also presented in the particulate phase (Shen et al., 2018; Wang et al., 2022; Tang et al., 2022). Recently, Xu et al. developed an enhanced method employing ultra-high-performance liquid chromatography and electrospray ionization tandem mass spectrometry (UHPLC-MS/MS) to simultaneously quantify 47 carbonyl compounds, showing that carbonyl compounds missed by the traditional method have a critical contribution to atmospheric photochemical pollution (Xu et al., 2023). Until now, the carbonyls detected by target analysis are only the tip of the iceberg, thus it is urgent to comprehensive identification of them in the atmosphere. In fact, this remains challenging due to the diversity and complexity of atmospheric chemical compositions.

In the past decade, non-targeted analysis methods based on ultra-high resolution mass spectrometry (UHRMS), such as Fourier transform ion cyclotron resonance mass spectrometry (FT-ICR-MS) and Orbitrap mass spectrometry, have been widely used for molecular characterization of soluble components in atmospheric aerosols from different sources (Song et al., 2022, 2019; Bianco et al., 2018a; Ma et al., 2022; Zeng et al., 2020; Yue et al., 2022). Based on non-targeted analysis, our understanding of the molecular composition and chemical diversity of atmospheric organic aerosols has greatly improved (Tang et al., 2022; Li et al., 2023a; Jiang et al., 2021). However, only molecular formulas in organic matter (OM) can be obtained by direct UHRMS detection, and the assessment of molecular characteristic groups is empirically dependent on their elementary composition. Chemical derivatization is a common technique used in targeted analysis for carbonyl



compounds (Liu et al., 2020; Baluha et al., 2013). Recently, Wang et al. employed Girard's reagent T derivatization with +ESI Orbitrap MS analysis to detect the carbonyls in atmospheric fine particulate matter ($PM_{2.5}$) (Wang et al., 2022).

However, all molecules containing three N atoms were assigned to derivative products, which may overestimate the contribution of carbonyls (Liu et al., 2020). In order to accurately and rapidly identify carbonyls from natural organic matter, we have recently developed a novel set of detection method and screening procedure based on *O*-(2,3,4,5,6-Pentafluorobenzyl)hydroxylamine (PFBHA) derivatization and ESI FT-ICR MS analysis (Yu et al., 2023a). This method provides the opportunity to reveal the molecular composition and chemical diversity of atmospheric carbonyl molecules.

Oxidative potential (OP) is one of the most widely used indicators to assess the atmospheric health hazards of $PM_{2.5}$. Previous studies have reported the important role of water-soluble organic matter (WSOM), such as quinones and humic-like substances (HULIS), in determining the OP of atmospheric aerosols (Jiang and Jang, 2018; Verma et al., 2012). Due to the chemical diversity of WSOM, it is still unclear which fractions contributed to OP of atmospheric aerosols. Carbonyls are known as major electrophilic compounds in atmospheric WSOM (Huang et al., 2020; Graber and Rudich, 2006; Huo et al.,

2021). Previous study have reported the correlation between the theoretical electrophilicity of model carbonyl compounds and dithiothreitol (DTT) activities (Chen et al., 2019a). Therefore, it is expected that atmospheric carbonyls may have an important contribution to the OP of WSOM in $PM_{2.5}$. However, until now, the relationship between carbonyls and DTT activities in actual atmospheric samples has not been addressed, mainly due to methodological limitations.

For the above reasons, in this study, we selected three stations in downtown, suburban, and montane sites of Beijing to

collect atmospheric $PM_{2.5}$ samples for typical pollution processes in summer and winter. The carbonyl molecules and non-carbonyl molecules in WSOM extracted from $PM_{2.5}$ samples were studied by PFBHA derivatization and ESI FT-ICR MS analysis. DTT assay was employed to measure the OP of the collected samples, and the contribution of carbonyls to the OP of WSOM extracted from real atmospheric samples was investigated.

## 2 Methods and Materials

### 2.1 $PM_{2.5}$ sample collection and extraction

$PM_{2.5}$ samples were collected in 2022 during three periods: winter (January 19 to February 3), Winter Olympics (February 4 to February 20), and summer (July 10 to July 27). The sampling sites were the monitoring sites set up by Beijing Urban Ecosystem Research Station, including a downtown site at the Beijing Teaching Botanical Garden (39°88'N, 116°44'E), a suburban site at Caiyu Town, Daxing District (39°67'N, 116°71'E), and a montane site at Mangshan National Forest Park

(40°28'N, 116°29'E). Samples were collected on pre-baked (550 °C, 6 h) quartz filters (Pallflex, 90 mm diameter) using medium-volume samplers with $PM_{2.5}$ cutting heads (Qingdao Laoying Corp, Qingdao, China) and then stored in a refrigerator at -20 °C. Particulate matter concentrations and meteorological data during the sampling period were provided by Beijing Urban Ecosystem Research Station. Detailed sampling information is recorded in Table S1 of the Supporting Information (SI).



The water-soluble fractions were extracted from PM$_{2.5}$ samples (each 1/4 quartz filter membrane) twice with ultrapure water (10 mL) and sonicated for 30 min each time. Then the solution was filtered through the polytetrafluoroethylene syringe filter (PTFE, 0.22 μm, Jinteng, Tianjin, China). The filtrate was further purified and desalted by solid phase extraction (SPE, Varian Bond Elute PPL cartridges) to obtain WSOM (Li et al., 2023b). The quartz filters after water extraction were dried and extracted twice (30 min each) with 5 mL of methanol by ultrasonication. Since inorganic salts and

metals were removed during ultrapure water extraction, the methanol extracted fraction is considered water-insoluble organic matter (WISOM) (Yue et al., 2022). The same extraction operation was performed for the travel blank. Detailed information on all chemicals used in the study is provided in Text S1 and more details about WSOM and WISOM extraction are provided in Text S2.

## 2.2 Derivatization of WSOM by PFBHA

Each WSOM sample was divided into three parts. One part was dried with N$_2$ and redissolved in ultrapure water for total organic carbon (TOC) analysis (Shimadzu 5000-A, Germany). Another part was dried with N$_2$ and redissolved in 50:50 methanol /water (v/v) for FT-ICR MS analysis. The third part was used for PFBHA derivatization before FT-ICR MS analysis. Briefly, PFBHA (300 mg L$^{-1}$) and WSOM samples (100 mg L$^{-1}$) were dissolved in a mixture of acetonitrile and water (70% and 30%, v/v) and then reacted in a shaker for 1 h and then incubated in a water bath at 50 °C for 12 h.

Subsequently, the samples were blown dry with N$_2$ and the solvent was replaced with acetonitrile and water (50% and 50%, v/v) and stored at -4°C.

## 2.3 ESI-FT-ICR MS analysis and carbonyl molecule identification

WSOM samples before and after derivatization were both detected using ESI-FT-ICR MS with the negative ion mode (15.0 T SolariX, Bruker. Billerica, MA, USA). The detection mass range was set to m/z 120-920, and mass peaks with a signal-to-

noise ratio > 4 are considered valid. Detailed analysis methods are described in the Supporting Information (Text S3).

Molecular formula assignment in the original WOSM was conducted following our previous study (Tang et al., 2022; Li et al., 2023b). For carbonyl molecule identification, a new automatic screening procedure developed in our group was employed (Yu et al., 2023a). Briefly, the mass difference of the reactant-product ion pair formed by the PFBHA derivatization should be 195.01074 (C$_7$H$_2$NF$_5$) or its multiples (C$_{14}$H$_4$N$_2$F$_{10}$, C$_{21}$H$_6$N$_3$F$_{15}$) according to the reaction equation

(Figure S1). The molecular formulas of carbonyls can be identified if the above mass difference was observed in the sample before and after PFBHA derivatization. As a result, carbonyl and non-carbonyl molecules in the original WOSM can be screened separately. Additionally, the molecular formulas of the control group (blank membrane samples) were subtracted for further analysis.

The molecular parameters such as double bond equivalence (DBE), modified aromaticity index (AImod), the nominal

carbon oxidation states (NOSC) (Wang et al., 2021),  and the average carbon oxidation state ($\overline{OS}_C$) (Kroll et al., 2011) of the





identified molecular formulas were calculated as documented in Text S3. The magnitude-weighted average parameters (e.g., DBEw) are the average of that compositional value in all formulas weighted by the relative intensity of each formula (Lv et al., 2016). The normalized molecular intensity was determined from the intensity of each molecular formula divided by the sum of the intensities of all identified molecular formulas in the sample (Lv et al., 2016).

## 2.4 DTT activity assay

OP was assessed based on the DTT activity assay (Dou et al., 2015; Wang et al., 2018; Lin and Yu, 2020). The DTT activity of the whole $PM_{2.5}$ samples was first assessed. The travel blanks and $PM_{2.5}$ samples (8 $cm^2$) were first ultrasonicated with 8 ml diethylenetriaminepentaacetic acid (DTPA, 1 mM, dissolved in 0.1 M phosphate buffer saline (PBS)) for 1 h, then the residual samples were added with 8 ml methanol for 1h. After filtering through 0.22μm PTFE membrane, PBS extract and methanol extract were each taken in a volume of 475 μL,mixed well, and then incubated with 50 μL DTT (2 mM, dissolved in 0.1 M PBS) in a cell incubator (37℃) for 0, 30, 60, 90, 120 min. Finally, 100 μL 5,5-dithiobis(2-nitrobenzoic acid) (DTNB, 1 mM, dissolved in 0.1 M PBS) was added and then the absorbance at 412 nm was measured by a microplate reader (SpectraMax iD3, Molecular Devices, USA).

For the WSOM and WISOM samples, 50uL of the samples were dissolved in 900uL of DTPA (1 mM, dissolved in 0.1 M PBS) and then incubated with 50 μL DTT and measured as described above. The whole experiment was conducted away from light and the consumption of DTT was controlled within 90%. Three repetitions were set for each sample and the experimental blank was deducted. As shown in Figure S2, the DTT concentrations varied linearly with time in this experiment, indicating a zero-level kinetic reaction, as reported in other studies (Ma et al., 2018; Chen et al., 2019b). As DTPA was added in this experiment to inhibit the consumption of DTT by metal ions, the measured values only reflect the response of OM to DTT (Lin and Yu, 2011). The volume-normalized DTT consumption rate ($DTT_v$), the mass-normalized DTT consumption rate ($DTT_m$), and the TOC-normalized DTT consumption rate ($DTT_{OC}$) are calculated by the following equations.

$$Volume-normalized\ DTT\ consumption\ rate\ (DTT_v)\ = \frac{R_{DTT}(\%) \times n_{DTT}(pmol)}{t(min) \times Air\ volumn(m^3)}$$

where $R_{DTT}$ (%) represents the ratio of DTT consumed in reaction time,$n_{DTT}$ is the amount of initial DTT, t is the reaction time, and Air volume ($m^3$) is the sampling volume corresponding to the extracted $PM_{2.5}$ samples.

$$PM_{2.5}\ Mass-normalized\ DTT\ consumption\ rate\ (DTT_m) = \frac{DTT_V(pmol\ min^{-1}\ m^{-3})}{PM_{2.5}\ mass\ concentration\ (\mu g\ m^{-3})}$$

$$TOC-normalized\ DTT\ consumption\ rate\ (DTT_{oc}) = \frac{DTT_V(pmol\ min^{-1}\ m^{-3})}{TOC\ concentration\ (\mu gC\ m^{-3})}$$



### 2.5 Reduction of the carbonyl group by Sodium borohydride (NaBH₄)

$NaBH_4$ has been reported to irreversibly reduce ketones/aldehyde to alcohols (Baluha et al., 2013), so that carbonyl

molecules in samples can be reduced by $NaBH_4$ treatment (Phillips and Smith, 2014; Baluha et al., 2013). Here, a standard humic substance Suwannee River natural organic matter (SRNOM) obtained from the International Humic Substances Society (IHSS) and a diesel soot sample collected from heavy-duty diesel vehicles were used to study the changes of carbonyls and DTT activity before and after $NaBH_4$ reduction. WSOM of diesel soot ($WSOM_{ds}$) was extracted with ultrapure water twice (10 ml, 30 min each) by ultrasonic extraction and filtered through the 0.22 µm filter membrane. Then the

SRNOM and $WSOM_{ds}$ were treated by $NaBH_4$ according to previous studies with minor modifications (Phillips and Smith, 2014, 2015). Briefly, the samples were deoxidized with $N_2$ for 15 min, then 10 times the sample mass of $NaBH_4$ was added and the reaction was performed in the dark for 12h, and the specific steps were recorded in Text S4 and Figure S3. Afterwards, the inorganic salts in the treated SRNOM and diesel soot samples, including unreacted $NaBH_4$ and the boron salts generated after the reaction, were removed by SPE (Text S2) (Lv et al., 2016). Then, the desalted samples of SRNOM

and diesel soot were further subjected to carbonyl derivatization, FT-ICR MS analysis, and DTT activity assay.

### 2.6 Statistical Analysis

The Kruskal-Wallis tests in IBM SPSS Statistics 25 were used to perform the significance test of difference for more than two groups of samples. The pro-oxidative carbonyls were screened stepwise by Spearman correlation analysis, correcting false discovery rates (FDR), and partial least squares regression (PLSR) modelling (Zhang et al., 2023). Spearman

correlation tests and FDR correction were performed using the packages "rcorr" and "fdrtool" in R v.3.6.2, respectively. The PLSR model, principal component analysis (PCA) model, and orthogonal partial least squares-discriminant analysis (OPLS-DA) model (Papazian et al., 2022) were performed and validated by SIMCA 14.1 software. Detailed description and model validation is recorded in Text S5 and Figure S4-6.

## 3 Results and discussion

### 3.1 The organic molecular composition of WSOM in PM$_{2.5}$

According to the daily average PM$_{2.5}$ concentrations at each site during the sampling periods (Figure S7), air quality is worse in winter (3-146 µg m$^{-3}$) than in summer (2-64 µg m$^{-3}$), but it has improved obviously during the Winter Olympics (5-43 µg m$^{-3}$). Based on the PM$_{2.5}$ concentration levels (> 80%, 40%-60%, and < 20%) in three periods, a total of 27 representative PM$_{2.5}$ samples (Table S1) were selected to further investigate the composition of organic molecules in WSOM by ESI FT-

ICR MS. The number of identified molecules in single sample ranged from 7369 to 10401. It seems that even in very clean air, WSOM extracted from PM$_{2.5}$ contained more than 7000 molecules. Totally, 29,203 unique molecules were identified in all samples, of which 5,147 were carbonyl molecules and 24,056 were non-carbonyl molecules (Figure 1a). The number of



carbonyls ranged from 11% to 22% in each sample, whereas the normalized molecular intensity of these carbonyls was higher (20%-35%), likely due to the higher concentration levels of carbonyl molecules than non-carbonyl molecules.

Significantly more carbonyl molecules were detected in winter than in summer (Kruskal-Wallis tests, $p < 0.05$), while non-carbonyl molecules detected in winter and summer showed no significant difference (Figure 1b), probably because more carbonyl compounds tended to be released into the gas phase in summer due to higher temperature (Qian et al., 2019b; Shen et al., 2018). At the same time, the fuel combustion emissions in winter may release more carbonyl molecules into the air (Fu et al., 2008).

The magnitude-weighted average parameters such as molecular weight and elemental ratio of organic molecules in each sample are summarized in Figure 1c. Compared to non-carbonyl molecules, carbonyl molecules generally have lower molecular weight ($MW_w = 374 \pm 14$), $DBE_w$ ($6.63 \pm 0.89$), and $AImod_w$ ($0.19 \pm 0.08$ values, but higher $O/C_w$ ($0.60 \pm 0.04$), $H/C_w$ ($1.36 \pm 0.11$) and $NOSC_w$ ($-0.01 \pm 0.04$), indicating carbonyl group mainly presented in smaller organic molecules with low aromaticity and high oxidation degree.

All the detected molecular formulas were classified into seven compound groups based on their O/C and H/C (Bianco et al., 2018b), and the proportion of molecules in each group was estimated based on the normalized molecular intensities (Figures S8 and S9). Lignins/carboxylic-rich alicyclic molecules (CRAMs)-like and proteins-like molecules are the most dominant components in both carbonyl and non-carbonyl molecules. Compared to non-carbonyl molecules, carbonyl molecules showed a significant decrease in lipids-like, lignins/CRAMs-like, unsaturated hydrocarbons-like ($p < 0.01$)
compounds, but a significant increase in carbohydrates-like and tannins-like compounds ($p < 0.001$).

    Additionally, molecular formulas can be categorized into four groups based on elemental composition and their proportions in carbonyl, non-carbonyl, and all molecules were estimated based on the normalized molecular intensities (Figures S10 and S11). On average, carbonyl molecules were primarily composed of CHO ($30.7\% \pm 10.7\%$) and CHOS ($30.1\% \pm 9.7\%$), while non-carbonyl molecules were predominantly comprised of CHON ($43.4\% \pm 4.5\%$) and CHO ($26.4\%$
$\pm 5.8\%$). Particularly, no obvious difference in CHO molecules was observed between carbonyl and non-carbonyl molecules, whereas the percentages of CHOS and CHON molecules displayed significant differences ($p < 0.001$) between carbonyl and non-carbonyl molecules, indicating a substantial contribution of carbonyl groups to sulfur-containing organic compounds.

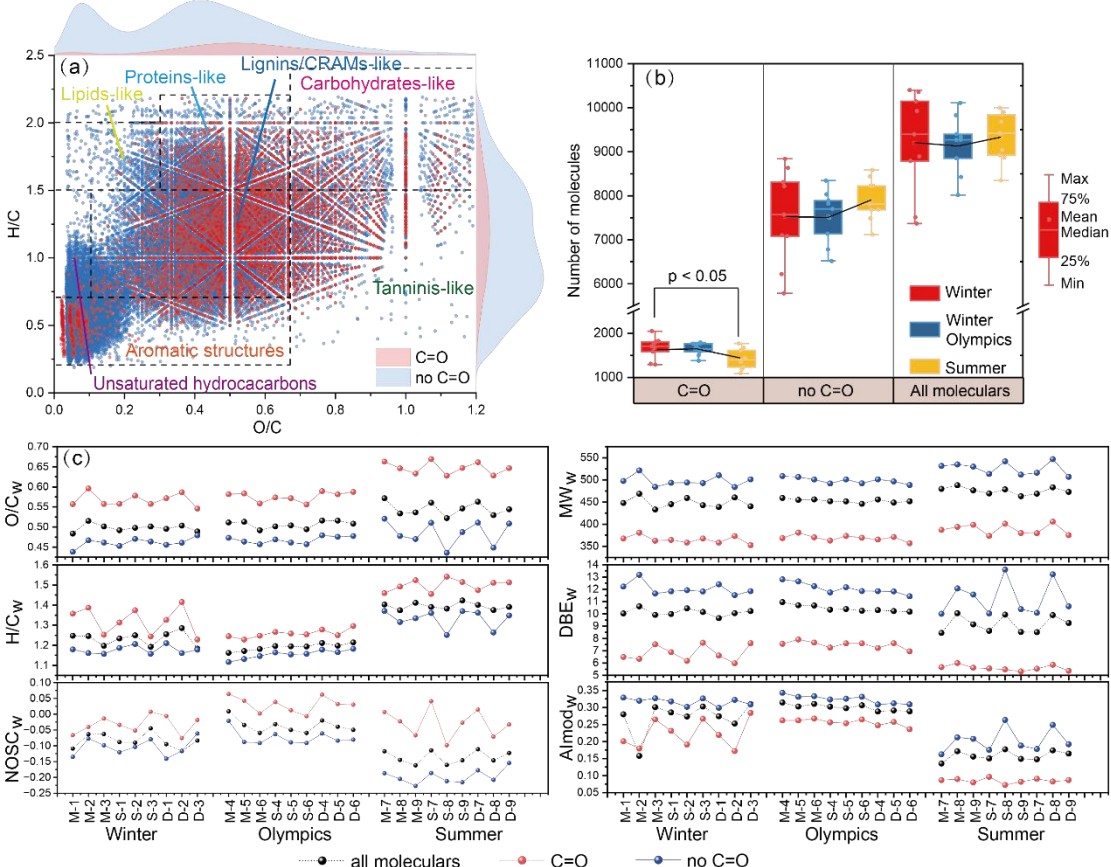

**Figure 1: The molecular composition of WSOM in PM2.5 samples was analyzed by FT-ICR MS. (a) Van Krevelen diagram of**
**carbonyl (C=O) and non-carbonyl (no C=O) molecules in all samples. (b) The distribution of total carbonyl and non-carbonyl molecules was detected in winter, Winter Olympics, and summer. (c) The magnitude-weighted average of O/C, H/C, the nominal oxidation states of carbon (NOSC), molecular weight (MW), double bond equivalence (DBE), and modified aromaticity index (AImod) of the samples. PM2.5 samples were shown as from the montane (M), suburban (S), and downtown site (D) in the winter (1-3), Winter Olympics (4-6), and summer (7-9) samples.**




## 3.2 Contributions of different organic fractions to the oxidative potential of $PM_{2.5}$

DTT activity is a commonly used cell-free assay to detect the OP of atmospheric samples (Bates et al., 2019; He and Zhang, 2022), wherein $DTT_v$ can reflect the DTT activity caused by $PM_{2.5}$ exposure, and $DTT_m$ can reflect the intrinsic DTT activity of the contaminants in $PM_{2.5}$ (Ma et al., 2018; Chen et al., 2019b; Daellenbach et al., 2020). Figure 2a illustrates the

variation of OP levels of $PM_{2.5}$ samples. The average $DTT_v$ of $PM_{2.5}$ samples in this study was $164 \pm 171$ pmol min$^{-1}$ m$^{-3}$, much lower than that reported for Beijing ($5.8 \pm 7.1$ nmol min$^{-1}$ m$^{-3}$) (Yu et al., 2022) and Tianjin($6.8 \pm 3.4$ nmol min$^{-1}$ m$^{-3}$) (Liu et al., 2018), but in the same order of magnitude as the other Chinese cities (Shanghai: 0.13 nmol min$^{-1}$ m$^{-3}$; Xi'an: 0.51 nmol min$^{-1}$ m$^{-3}$; Hangzhou: 0.62 nmol min$^{-1}$ m$^{-3}$) (Lyu et al., 2018; Chen et al., 2019b; Wang et al., 2019).

Similar to $PM_{2.5}$ concentration, samples collected in summer displayed the lowest $DTT_v$ (8-90 pmol min$^{-1}$ m$^{-3}$), followed

by Winter Olympics (32-660 pmol min$^{-1}$ m$^{-3}$) and winter (43-741 pmol min$^{-1}$ m$^{-3}$). Both $DTT_v$ and $DTT_m$ were significantly higher in winter than in summer ($p < 0.001$), indicating that both of the contributions of total organic aerosol and organic aerosol per unit mass to OP are higher in winter, and therefore the overall OP risk is higher in winter. In addition, although the $DTT_v$ and $DTT_m$ values for the Winter Olympics were slightly lower than those in winter, they were still much higher than those in summer, suggesting that the regulation of neighboring industrial emissions and traffic during the Winter

Olympics had a limited contribution to the reduction of atmospheric OP. Hence winter heating and meteorological conditions may be the main influences on the OP of $PM_{2.5}$. As shown in Figure S12, no significant difference in $DTT_v$ was observed between regions. However, the $DTT_m$ of suburban sites is significantly higher than that of urban sites in winter, and the opposite is true in summer ($p < 0.05$). This is likely due to the differences in aerosol sources in different seasons.

The DTT activities of WSOM and WISOM fractions for each sample were detected as well, results showed that the

contribution of WSOM to $DTT_v$ (51%-84%) was much higher than that of WISOM (Figure 2b), indicating water extractable compounds have a greater contribution to the OP of $PM_{2.5}$. Meanwhile, we noticed that the seasonal variation of carbonyl molecules (Figure 1b) was consistent with the $DTT_{OC}$ (Figure 2c), which was higher in winter than in summer. Further, the correlation between the $DTT_{OC}$ and organic molecules was investigated (Figure 2d). The results showed that the abundance of non-carbonyl molecules was negatively correlated with $DTT_{OC}$ (Pearson's r = -0.263), whereas the abundance of carbonyl

molecules was significantly positively correlated with $DTT_{OC}$ (Pearson's r = 0.55, $p < 0.01$), suggesting carbonyl molecules played a more important role in OP generation than non-carbonyl molecules. By using simulated organic aerosol samples, previous studies reported that the DTT activity is modulated by carbonyl-containing organic molecules such as quinone and peroxyacetyl nitrates (Jiang and Jang, 2018), and the unsaturated carbonyl compounds can cause toxicity by alkylating thiols in proteins (Han et al., 2020). Here, we observed the relationships between carbonyl molecules and DTT activities of real

aerosol samples for the first time, providing further evidence for the health hazards of carbonyl molecules in atmospheric environments.





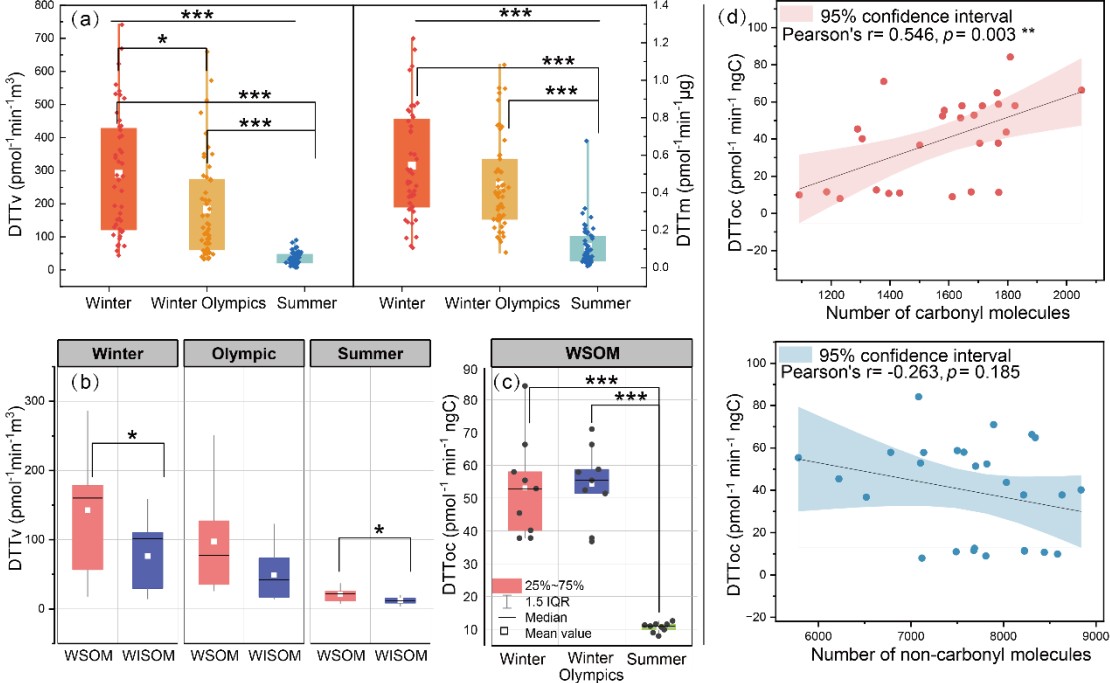

**Figure 2: DTT consumption rates of different fractions in PM₂.₅ and its correlation with organic molecules over Beijing in the**
**winter, Winter Olympics, and summer (Kruskal-Wallis tests, differences between groups were considered statistically significant**
**when $p < 0.05$, with $0.01 < p < 0.05$ marked by \*; $p < 0.01$ marked by \*\*; and $p < 0.001$ marked by \*\*\*). (a) Volume-normalized**
**DTT consumption rate (DTT$_v$) and mass-normalized DTT consumption rate (DTT$_m$) of organic matter in PM₂.₅ of the sampling**
**period. (b) DTT$_v$ of water-soluble organic matter (WSOM) and water-insoluble organic matter (WISOM) for representative**
**samples. (c) DTT consumption rate of WSOM normalized by TOC (DTT$_{OC}$). (d) Relationship between carbonyl and non-carbonyl**
**molecules and oxidative potential of WOSM.**

**3.3 Verification of the contribution of carbonyl molecules to the oxidative potential of organic aerosol**

To verify the contribution of carbonyl molecules to the OP of organic aerosol, NaBH₄ was employed to reduce carbonyls in
WSOM to alcohols (Phillips and Smith, 2015), and the changes of DTT activities before and after NaBH₄ treatment were
investigated. SRNOM obtained from IHSS and WSOM extracted from diesel soot (WSOM$_{ds}$) were used as representative
WSOM samples. FT-ICR MS results showed that SRNOM and WSOM$_{ds}$ contained 1367 and 1722 carbonyl molecules,
respectively (Figure 3a, 3b). Unlike SRNOM, the carbonyl molecules of WSOM$_{ds}$ had a higher number of CHON (48.1%),
likely due to the reaction between the high amount of nitrogen oxide (NOx) and VOC released from diesel vehicles. The
original OP of SRNOM and WSOM$_{ds}$ were $23.83 \pm 1.07$ pmol min⁻¹ µgC⁻¹ and $57.28 \pm 3.05$ pmol min⁻¹ µgC⁻¹. The higher
OP of WSOM$_{ds}$ was attributed to its higher contents of carbonyl molecules. After NaBH₄ treatment, 62.9% (619) and 64.1%
(1103) of carbonyl molecules in SRNOM and WSOM$_{ds}$ were removed respectively, leaving 24.5% (335) and 37.1% (619) of
carbonyl molecules that cannot be removed (Figure 3c, 3d). This is the first time to evaluate the effect of NaBH₄ on the
reduction of carbonyls in WSOM instead of model compounds because the new detection method was used. As predicted,
DTT activities of the SRNOM and WSOM$_{ds}$ reduced 41.3% and 77.2% after NaBH₄ treatment (Figure 3e, 3f), which was




consistent with the reduction of carbonyl molecules in two samples by NaBH$_4$ treatments. These results further confirmed the important role of carbonyl molecules in determining the OP of organic aerosol.

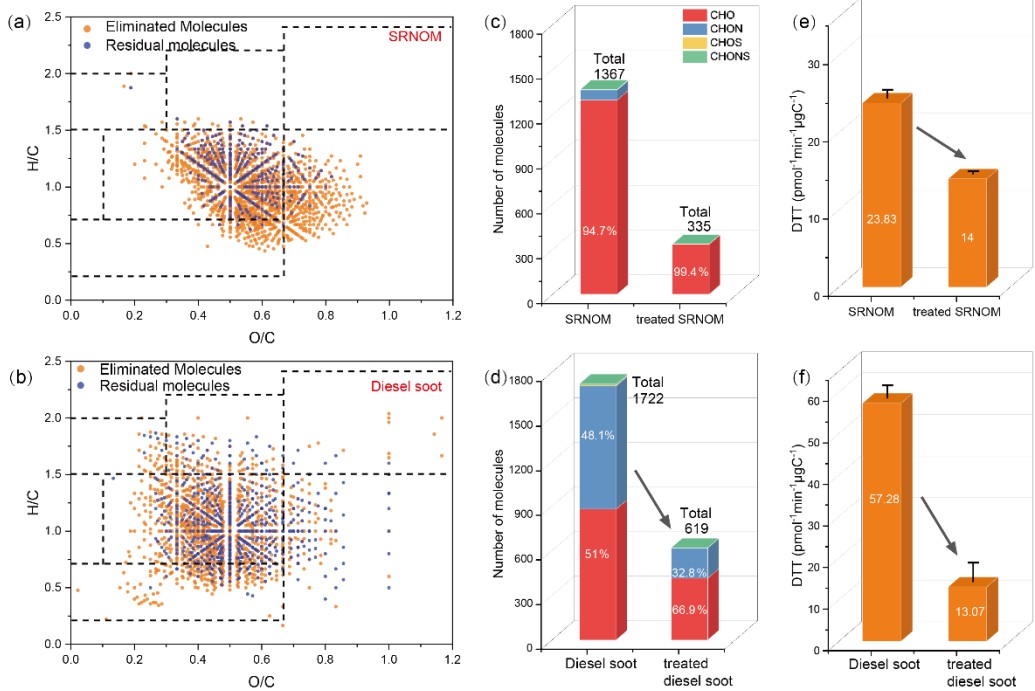

**Figure 3: Van Krevelen diagrams of carbonyl molecules in SRNOM (a) and WSOM extracted from diesel soot (b), with orange points representing carbonyl molecules eliminated by NaBH$_4$ and blue points representing residual carbonyl molecules. Changes in carbonyl molecule composition of SRNOM (c) and WSOM extracted from diesel soot (d) before and after NaBH$_4$ treatment. Changes in DTT$_{OC}$ before and after NaBH$_4$ treatment of SRNOM (e) and WSOM extracted from diesel soot (f).**

### 3.4 Molecular composition of carbonyls in PM$_{2.5}$

Because of the importance of carbonyl molecules in determining the OP of PM$_{2.5}$, molecular characteristics for carbonyls were further studied. In general, the DBE$_w$ and AImod$_w$ values of carbonyl molecules detected in winter were higher than those in summer, while the MW$_w$, O/C$_w$, and H/C$_w$ values of carbonyl molecules detected in winter were lower (Figure S13). By using cluster analysis of the samples based on the composition of carbonyl molecules (Figure 4a), two distinct clusters of summer and winter samples were successfully obtained. In one of the clusters, the samples collected during the strict control period of the Winter Olympics can be distinguished from the other winter samples. The proportions of carbonyl molecules in the seven compound groups were estimated based on the normalized molecular abundance. As shown in Figure S14, lignins/CRAMs-like carbonyl molecules were the most abundant components during winter (52.0% ± 9.4%) and Winter Olympics (57.6% ± 2.8%), and their abundance was significantly higher than that in summer (28.1% ± 3.5%) ($p < 0.01$). In summer, besides lignins/CRAMs-like carbonyl molecules, proteins-like carbonyl molecules (28.1% ± 5.7%) were also one



of the most dominant components. In addition, the abundance of carbohydrates-like and tannins-like was significantly higher in winter than in winter and Winter Olympics periods ($p < 0.05$).

The elementary composition of the carbonyl molecules was further analyzed. In general, the CHO-containing carbonyl molecules were distributed in regions with lower H/C values dominated by lignins/CRAMs-like molecules, while the sulfur-containing carbonyl molecules (CHOS, CHONS) were distributed in the region with higher H/C values dominated by

carbohydrates-like and proteins-like (Figure S15). Furthermore, CHOS-containing carbonyl molecules comprised the highest proportion during winter ($31.9\% \pm 9.9\%$) and summer ($36.8\% \pm 6.9\%$) (Figure 4b). In contrast, during the Winter Olympics, CHO occupied the highest position with $39.5\% \pm 6.1\%$ of the total, while CHOS accounted for only $21.5\% \pm 2.7\%$. This discrepancy may be attributed to control measures implemented during the Winter Olympics, such as limiting the operations of high-emission and high-pollution enterprises, which potentially led to a reduction in sulfur-containing compounds.

However, it is noteworthy that sulfur-containing carbonyls do not constitute the primary source of aerosol OP, as the $DTT_{OC}$ during the Winter Olympics remained comparable to that observed in other days (Figure 2c).

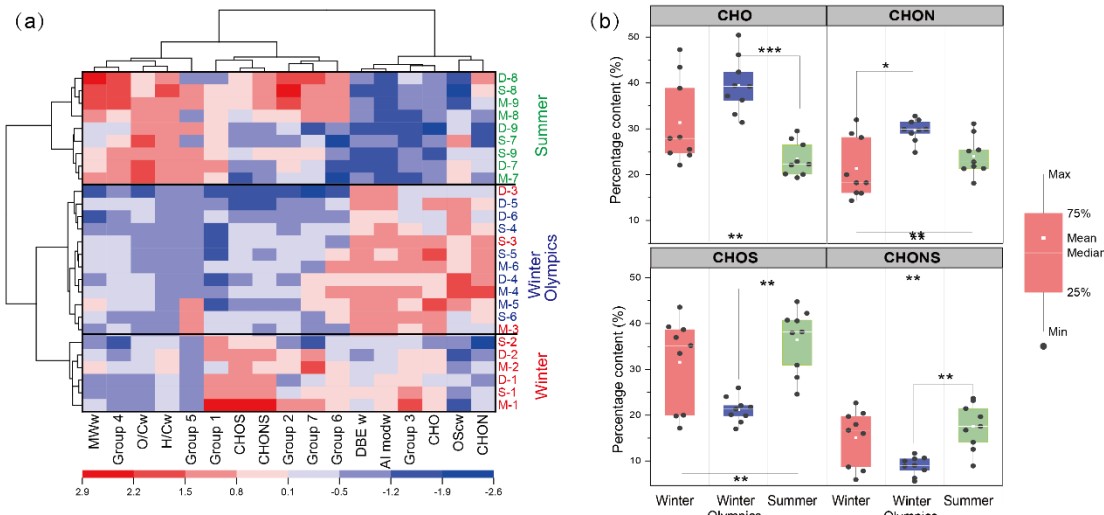

**Figure 4: Molecular composition and distribution characteristics of carbonyl molecules in WSOM of PM2.5 samples. (a) The heatmap shows the correlation between the carbonyl molecular characteristics and the samples collected from the winter (1-3), Winter Olympics (4-6), and summer (7-9) in the montane (M), suburban (S), and downtown site (D) (Group1: lipids-like, Group2:**
**proteins-like, Group3: lignins/CRAMs-like, Group4: carbohydrates-like, Group5: unsaturated hydrocarbons, Group6: aromatic structures, Group7: tannins-like). (b) The difference in the percentage distribution based on the normalized molecular intensities of the elementary composition (CHO, CHON, CHOS, CHONS) of carbonyl molecules in winter, Winter Olympics, and summer samples.**

### 3.5 Potential carbonyl markers of atmospheric oxidative stress

Carbonyl compounds with different structures should have different DTT activities. Spearman correlation matrix of the characteristic parameters of carbonyl molecules with $DTT_{OC}$ revealed that $DBE_w$ and $AImod_w$ were significantly positively correlated with $DTT_{OC}$, while $MW_w$, $O/C_w$, and $H/C_w$ were significantly negatively correlated with $DTT_{OC}$ (Figure S16). Especially, the number of aromatic carbonyl groups (AImod > 0.5) (Koch and Dittmar, 2006) in the samples showed a





higher correlation with the $DTT_{OC}$ (Spearman's r = 0.779, $p < 0.001$), suggesting more attention should be paid to the

aromatic carbonyl molecules.

In order to screen out potential carbonyl markers to indicate atmospheric oxidative stress (pro-oxidative carbonyls) from numerous carbonyl molecules, Spearman rank correlation and PLSR model analysis using the normalized intensities of carbonyl molecules with more than 50% detection rate and $DTT_{OC}$ were performed. The results showed that 380 carbonyl molecules positively correlated with $DTT_{OC}$ had a value of variable importance in projection scores (VIP) greater than 1.0,

indicating these carbonyl molecules had greater DTT activity, thus can be considered as typical pro-oxidative carbonyls. As shown in Figure 5a, these pro-oxidative carbonyls were mainly lignins/CRAMs-like (86%), and dominated by CHO (45.5%) and CHON (42.6%). The distribution region of molecules in the average carbon oxidation states - C number space has been used to explain the sources of organic aerosol (Kroll et al., 2011). According to Figure 5b, the identified pro-oxidative carbonyls were mainly sourced from fresh secondary organic aerosols. This suggested that the risk of atmospheric oxidative

stress mainly comes from secondary aerosol products and is therefore more difficult to reduce.

OPLS-DA model analysis was performed to further reveal the potential carbonyl markers in different seasons. Results showed that samples collected in winter and summer could be distinguished from each other by the pro-oxidative carbonyls (Figure S17). Then the molecules with VIP > 1 and had the highest model-correlation coefficients ($|p[corr]| > 0.5$) in the volcano plot were screened out as molecular markers in different seasons. Overall, 167 molecular markers in winter were

found, which were mainly composed of CHO elements and had significantly higher aromaticity (DBE=11 ± 2, AImod=0.47 ± 0.16) and molecular weight (MW=368 ± 65) compared to other pro-oxidative carbonyls (DBE=9 ± 2, AImod=0.42 ± 0.19, MW=329 ± 65). However, no pro-oxidative carbonyls were found in summer (Figure 5c) due to the low $DTT_{OC}$ of $PM_{2.5}$ collected in summer. These results suggested that aromatic secondary products containing carbonyl group produced from combustion by-products in winter are potential molecular markers of atmospheric oxidative stress.

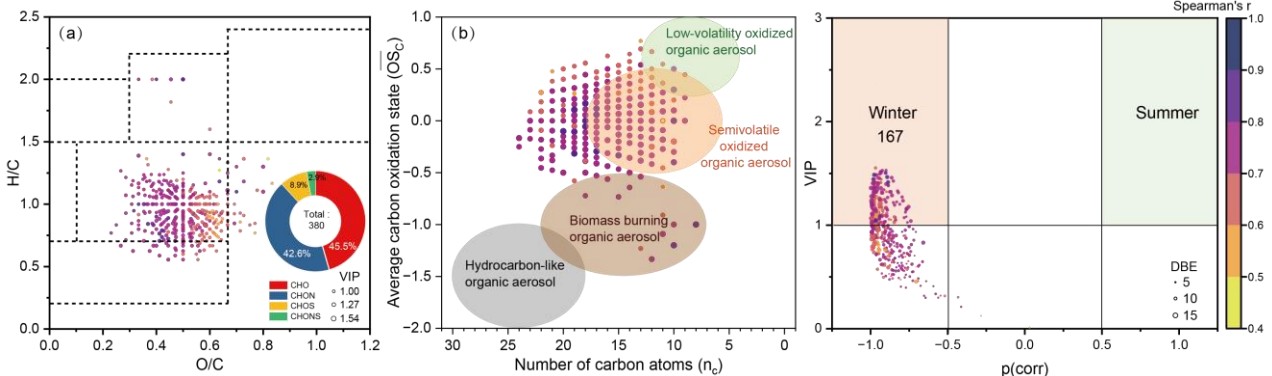

**Figure 5: (a) Van Krevelen diagram of pro-oxidative carbonyls. (b) The average carbon oxidation state-C number space of pro-oxidative carbonyls. The source shadings are marked referring to Kroll et al (Kroll et al., 2011). Semivolatile and low-volatility oxidized organic aerosol (SV-OOA and LV-OOA) correspond to 'fresh' and 'aged' secondary aerosol produced by multistep oxidation reactions (Jimenez et al., 2009). (c) From the OPLS-DA model, the contribution of each pro-oxidative carbonyls to winter versus summer in the volcano plot, showing seasonal variation markers with the highest model-correlation coefficients**

**($|p[corr]| > 0.5$) and variable importance in projection scores (VIP > 1.0). Features are colored by their Spearman correlation coefficients.**





## 4 Conclusions

PM$_{2.5}$ contains a large number of organic compounds, which have potential harm to the ecological environment and human health. In this study, we investigated the abundance and molecular diversity of carbonyl and non-carbonyl molecules in

WSOM from PM$_{2.5}$ using FT-ICR MS analysis with chemical derivatization, taking Beijing as a research area. The method expanded the application of FT-ICR MS in identifying compounds with specific functional groups in the atmosphere, and it can be used to study other forms of atmospheric organic matter, such as brown carbon, cloud and mist, fumes, and automobile exhaust. At the same time, the molecular composition and diversity of carbonyl molecules in PM$_{2.5}$ in other regions are also worth studying. Therefore, a more comprehensive understanding of the composition and sources of

carbonyls in aerosols can be obtained.

In addition, this study found that carbonyls, especially aromatic carbonyl molecules, played a more important role in atmospheric oxidative stress, suggesting more attention should be paid to the possible health hazards posed by atmospheric carbonyls. There are many ways to produce atmospheric carbonyls in the real environment, except for direct emissions such as biomass combustion, atmospheric secondary oxidation reactions play a substantial  role in carbonyls generation. Various

small carbonyls are water-soluble products of ozone- and hydroxyl radical-induced oxidation of biological-derived hydrocarbons (i.e., isoprene (Surratt et al., 2006; Nguyen et al., 2011) and terpenes (Docherty et al., 2005)) and anthropogenic-derived hydrocarbons (i.e., aromatics, (Kalberer et al., 2004b) alkene and cycloalkene (Gao et al., 2004)). Further, these carbonyls, such as acetaldehyde, acetone, glyoxal, methylglyoxal, and methyl vinyl ketone, could form oligomers via OH radical-induced oxidation (Lim et al., 2010; Renard et al., 2013) or non-radical reactions (hemiacetal-

acetal (Loeffler et al., 2006), hydroxyaldehyde (Yasmeen et al., 2010; Chan et al., 2013), and esterification (Altieri et al., 2008)). For example, isoprene oxidized by hydroxyl radicals under light conditions produces 2-methylglyceric acid and glycolaldehyde, which further undergo condensation and addition oligomerization (Nguyen et al., 2011). Therefore, secondary organic aerosol generation deserves great attention as it may be the main source of carbonyl compounds in PM$_{2.5}$.

As we know, anthropogenic emission is an important source of PM$_{2.5}$, thus the pollution of PM$_{2.5}$ can be controlled by

emission limitation. In this study, carbonyl molecules in PM$_{2.5}$ during the 2022 Beijing Winter Olympics were compared with other days. Therefore, the effects of emission limitation on the OP and carbonyl molecules of atmospheric aerosols were studied. Unlike PM$_{2.5}$ pollution, no significant decrease in atmospheric carbonyls during the Winter Olympics was observed, and the OP of PM$_{2.5}$ in this period did not reduce as well. Therefore, the control of atmospheric carbonyl pollution is a more challenging task than that of PM$_{2.5}$ pollution, and more scientific measures are needed to reduce the health hazards

caused by atmospheric oxidation stress.





**Data availability.**

The data used in this study are available from the corresponding author upon request (jtlv@rcees.ac.cn).

**Author contribution.**

F. F. Li: methodology, formal analysis, investigation, visualization, writing – original graft. S. S. Tang: validation, investigation, methodology. J. T. Lv: conceptualization, methodology, formal analysis, writing - review & editing, supervision, funding acquisition. S. Y. Yu: methodology. X. Sun: investigation. D. Cao: methodology, validation. Y. W. Wang: conceptualization, writing – review & editing, supervision, funding acquisition. G. B. Jiang: supervision, validation.

**Disclaimer**

Publisher's note: Copernicus Publications remains neutral with regard to jurisdictional claims in published maps and institutional affiliations.

**Competing interests.**

The authors declare that they have no conflict of interest.

**Acknowledgements.**

We thank the help of the Beijing Urban Ecosystem Research Station in sampling and data sharing.

**Financial support.**

This research was jointly supported by the National Natural Science Foundation of China (22021003, 22106031), the K.C. Wong 394 Education Foundation of China (GJTD-2020-03), and the Youth Innovation Promotion Association of the Chinese Academy of Sciences (2020044).

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
