# Peer review of "Critical contribution of chemically diverse carbonyl molecules to the oxidative potential of atmospheric aerosols"

_EGUsphere, 2024_

## Author Comment (AC1)

**Point-by-Point Responses to the Comments from Reviewers**

Manuscript ID: egusphere-2024-37

Manuscript Title: "Critical contribution of chemically diverse carbonyl molecules to the oxidative potential of atmospheric aerosols"

The corresponding authors: Prof. Jitao Lv and Prof. Yawei Wang

**Responses to the Comments from Anonymous Referee 1:**

***Comments:***

*The authors Li et al describe a series of analyses on the oxidative potential (OP) and molecular composition of filter-collected ambient particles as well as the inter-relation between these two measured qualities. The experiments are well executed and provide interesting insights on both ambient organic aerosol (OA) composition as well as the compounds affecting OP within OA and the results will be useful in informing how we understand the health effects of ambient aerosol. I have several comments on the methodologies employed and some of the conclusions drawn, but I believe that after these comments are addressed the manuscript will likely be suitable for publication.*

**Response:** We sincerely appreciate your positive comments. We have made careful revisions according to your comments and provided a point-by-point explanation for each comment.

***Section 2.2:***

*1. Why were different solvent conditions (methanol vs ACN) used for WSOM samples that were directly analyzed by FT-ICR MS and those that were first derivatized? Additionally, prior work (e.g., Chen et al., 2022) has observed methanol (but not ACN) extraction to induce chemical reaction in some carbonyl-containing molecules; in the work of Chen et al., this was observed for the carbonyl-containing molecules phthalic and maleic anhydride. The full impacts of methanol-induced reactions during extraction in the present work is not immediately clear to me, but this possibility should be discussed.*

**Response:** Thanks for your insightful comment. For FT-ICR MS analysis, the extracted samples were re-dissolved in 50% MeOH and analyzed by ESI-FT-ICR MS immediately. We used MeOH rather than ACN as a solvent for FT-ICR MS analysis because some SOA constituents are more soluble in MeOH (Daellenbach et al., 2019; Jiang et al., 2022; Song et al., 2022).

However, during the carbonyl derivatization reaction, we utilized ACN as the solvent according to previous studies (Liu et al., 2020; Alaimo et al., 2021). We have read Chen's paper carefully and made revisions to discuss why ACN rather than MeOH was used as a solvent in the derivatization procedure. It is regrettable that we did not specifically focus on whether using 50% MeOH as a solvent for FT-ICR MS analysis can induce artifacts for WOSM extracted from $PM_{2.5}$. This is a subject worthy of future study.

Revisions at lines 120-121: Here, ACN was used as a solvent to avoid the formation of potential solvent artifacts by reacting MeOH with conjugated carbonyl groups during the derivatization reaction (Chen et al., 2022).

**Reference:**

Daellenbach, K. R., Kourtchev, I., Vogel, A. L., Bruns, E. A., Jiang, J., Petäjä, T., Jaffrezo, J.-L., Aksoyoglu, S., Kalberer, M., Baltensperger, U., El Haddad, I., and Prévôt, A. S. H.: Impact of anthropogenic and biogenic sources on the seasonal variation in the molecular composition of urban organic aerosols: a field and laboratory study using ultra-high-resolution mass spectrometry, Atmospheric Chemistry and Physics, 19, 5973–5991, https://doi.org/10.5194/acp-19-5973-2019, 2019.

Jiang, H., Li, J., Tang, J., Cui, M., Zhao, S., Mo, Y., Tian, C., Zhang, X., Jiang, B., Liao, Y., Chen, Y., and Zhang, G.: Molecular characteristics, sources, and formation pathways of organosulfur compounds in ambient aerosol in Guangzhou, South China, Atmospheric Chemistry and Physics, 22, 6919–6935, https://doi.org/10.5194/acp-22-6919-2022, 2022.

Song, J., Li, M., Zou, C., Cao, T., Fan, X., Jiang, B., Yu, Z., Jia, W., and Peng, P.: Molecular Characterization of Nitrogen-Containing Compounds in Humic-like Substances Emitted from Biomass Burning and Coal Combustion, Environ. Sci. Technol., 56, 119–130,

https://doi.org/10.1021/acs.est.1c04451, 2022.

Liu, X., Liu, R., Zhu, B., Ruan, T., and Jiang, G.: Characterization of Carbonyl Disinfection By-Products During Ozonation, Chlorination, and Chloramination of Dissolved Organic Matters, Environ. Sci. Technol., 54, 2218–2227, https://doi.org/10.1021/acs.est.9b04875, 2020.

Alaimo, C. P., Li, Y., Green, P. G., Kleeman, M. J., and Young, T. M.: Diversity of Carbonyl Compounds in Biogas and Natural Gas Revealed Using High-Resolution Mass Spectrometry and Nontarget Analysis, Environ. Sci. Technol., 55, 12809–12817, https://doi.org/10.1021/acs.est.1c01646, 2021.

2. *Can the authors comment on the expected efficiency of the carbonyl-PFBHA reaction? From a quick look through the literature the extent of reaction seems to have some variability across different reactants and conditions. I realize that myriad molecules make up OA/WSOM and an evaluation of all is impossible, but some discussion or analysis of the expected extent of reaction with a small subset (for example, carbonyl molecules identified in prior work as contributing to OP, lines 251-254) would be useful.*

**Response:** Thanks for your good comments. We have collected standard substances of six model carbonyl molecules with different structures (quinones, aldehydes, ketones, and acids) to evaluate the efficiency of the carbonyl-PFBHA reaction, including 9,10-anthraquinone, 5-Hydroxy-1,4-naphthoquinone, Benzoylformic acid, trans-Cinnamic acid, Syringaldehyde, Coumarin. These substances were chosen based on prior literature highlighting their significance in OP. Following the experimental conditions described in Section 2.2, we conducted carbonyl derivatization experiments with a mass ratio of PFBHA to all carbonyls set at 3:1. The concentrations of each carbonyl molecule were varied in gradients of 100, 50, and 10 $\mu g\ L^{-1}$. Subsequently, we qualitatively assessed the detection efficiency of the derivatized products in different treatments. As shown in Figure R1 (Figure S1 in SI) six model compounds and their derivatized products were detected in all treatments. These derivatized products displayed considerable or higher S/N than original model carbonyls, indicating high derivatization and detection efficiencies of derivatized products using our method. In fact, derivatized products were only used to track carbonyls in original sample in this study. We focused on if the derivative product can be detected by the FT-ICR MS; if it is detected, we trace it back to the molecules in original sample through the carbonyl-PFBHA reaction pair, rather than directly using the derivative product for semi-quantitative analysis.

Detailed results of model carbonyl molecules have been provided in lines 121-126 and Figure S1 of the revised manuscript.

[Figure]

**Figure R1. Mass spectra of 9,10-anthraquinone, 5-hydroxy-1,4-naphthoquinone, benzoylformic acid, trans-Cinnamic acid, syringaldehyde, coumarin and their PFBHA derivatives in the sample derivatized by PFBHA.**

***Section 2.3***

3. *To what extent does the upper limit of the mass range impact the analysis? For example, a molecule with one carbonyl group and exact mass > ~725 would be detected in the non-derivatized sample but the derivatized analyte would be above the mass range and not detected, potentially leading to undercounting the number of carbonyl molecules and affecting the quantities reported in Figure 1 and the associated discussion on molecular characteristics. Relatedly, what proportion of detected carbonyls are singly vs doubly vs triply derivatized? My understanding is that it's not uncommon for molecular components of OA to contain multiple carbonyl groups, particularly molecules derived from aromatic-containing precursors, and it appears that such molecules may only be detected (as carbonyl-containing) when the underivatized molecule is relatively much smaller than many other analytes.*

**Response:** Thanks for your insightful comments. It is right that the upper limit of the mass range may impact the analysis, because the reactant-product ion pair formed by the PFBHA derivatization should have a mass difference of 195.01074 ($C_7H_2NF_5$) for monocarbonyl molecule, 390.02148 ($C_{14}H_4N_2F_{10}$) for dicarbonyl molecule, and tricarbonyl molecule 585.03222 ($C_{21}H_6N_3F_{15}$). We tested the upper limit of the derivative sample by setting the mass range to m/z = 99-2000. As shown in Figure R2, the intensities of mass peaks above 900 Da were low with S/N below 4. Considering the calibration range of sodium formate internal standard, we selected the molecular ion peaks in mass range of m/z 122-928 for further molecular formula assignment. Among all carbonyl molecules detected in this study, monocarbonyl molecules accounted for 48%-90% (997-1853), dicarbonyl molecules accounted for 5%-18% (101-360), and tricarbonyl molecules accounted for less than 1% (less than 20). Because of the insignificant amount of tricarbonyl molecules, the mass range of 122-928 Da did not affect the detection of carbonyl molecules in this study. The related revisions are provided at lines 129-131 and Table S2.

[Figure]

**Figure R2. Broadband ESI-FT-ICR-MS spectrum of WSOM sample before and after PFBHA derivatization, the mass range was set to (a) 122-928 Da and (b) 99-2000 Da.**

4. *Is the automatic screening procedure referenced from Yu et al. (2023a) also described in a peer-reviewed publication? If so, please reference it here, and if not, please add more detail to the current manuscript.*

**Response:** Thanks for your suggestion. The detailed screening process of the automated screening program was added at line 135, Text S4 and Figure S2.

***Section 3***

5. *Lines 205-206: I believe more context is needed around the seven categories overlayed on the Van Krevelen diagrams. These categories were originally developed following analysis of dissolved organic matter rather than OA, so the delineations most relevant for DOM may not be the most useful for OA, with its own unique combination of primary and secondary organic molecules. A discussion of the category definitions and molecular characteristics would provide more clarity to the aerosol community.*

**Response:** Thanks for your comments. We agree that this classification was originally based on some conventional molecular composition in DOM extracted from soil and water. We have rearranged the molecular classification according to previous studies of atmospheric aerosol (Su et al., 2021; Jiang et al., 2022), mainly based on the degree of aromatization and alkylation, rather than on the particular chemical composition. In the van Krevelen (VK) diagrams, all the detected molecular formulas were divided into five categories, including polycyclic aromatic-like molecules (AImod > 0.67), highly aromatic-like molecules (0.5 < AImod ≤ 0.67), highly unsaturated-like molecules (AImod ≤ 0.5 and H/C < 1.5), unsaturated aliphatic-like molecules (1.5 ≤

H/C < 2.0), and saturated-like molecules (H/C ≥ 2.0 or O/C ≥ 0.8). The revised category definitions and molecular characteristics are presented at lines 145-149 and all discussions involving categorization have been revised.

**Reference:**

Su, S., Xie, Q., Lang, Y., Cao, D., Xu, Y., Chen, J., Chen, S., Hu, W., Qi, Y., Pan, X., Sun, Y., Wang, Z., Liu, C.-Q., Jiang, G., and Fu, P.: High Molecular Diversity of Organic Nitrogen in Urban Snow in North China, Environ. Sci. Technol., 55, 4344–4356, https://doi.org/10.1021/acs.est.0c06851, 2021.

Jiang, H., Li, J., Tang, J., Cui, M., Zhao, S., Mo, Y., Tian, C., Zhang, X., Jiang, B., Liao, Y., Chen, Y., and Zhang, G.: Molecular characteristics, sources, and formation pathways of organosulfur compounds in ambient aerosol in Guangzhou, South China, Atmospheric Chemistry and Physics, 22, 6919–6935, https://doi.org/10.5194/acp-22-6919-2022, 2022.

6. *Lines 248-251: It's not clear to me from the text and axis labels in Figure 2c,d whether the "abundance" of carbonyl molecules used for correlation analysis is just the number of carbonyl molecules (as implied from the labels in Figure 2c,d) or an abundance where each carbonyl signal is weighted by its signal intensity (which may make more sense).*

**Response:** Thanks for your great suggestion. In Figure 2c,d, the abundance of carbonyl molecules is exactly the number of detected carbonyl molecules, in order to avoid ambiguity we have revised "abundance" to "number" of carbonyl molecules in the related contents. Both the number and the number proportion of carbonyl molecules were positively correlated with $DTT_{OC}$ (Pearson's r = 0.55, $p$ < 0.01; and Pearson's r = 0.63, $p$ < 0.001), whereas an insignificant or negative correlation was observed between those of non-carbonyls and $DTT_{OC}$ (Pearson's r = -0.26, $p$ = 0.185; and Pearson's r = -0.63, $p$ < 0.001). We also performed a correlation analysis between $DTT_{OC}$ and the normalized molecular intensities of carbonyls or non-carbonyls of each sample. Insignificant correlation was observed between the normalized molecular intensities of all carbonyls or non-carbonyls and $DTT_{OC}$. However, a significant positive correlation between the normalized molecular intensities of carbonyls and $DTT_{OC}$ was obtained (Pearson's r = 0.56, $p$ < 0.01) when sulfur-containing molecules were excluded (Figure R3 c). Sulfur-containing molecules in aerosols often displayed extremely high ionization efficiencies, leading to unreasonably high intensities of S-containing molecules detected by FT-ICR MS (Xie et al., 2022). When considering normalized molecular intensities in correlation analysis, it is more reasonable to exclude sulfur-containing molecules. The related

revision and discussion have been added at lines 268-271 and Figure S14 in the revised manuscript.

In fact, it is a great challenge to quantification of mass data obtained by FT-ICR MS, as the peak intensity in a mass spectrum is not only determined by the initial concentration but also by the ionization efficiency of molecules. Using statistics based on the number of molecules can avoid the effect of ionization efficiency, but the concentration difference between molecules is ignored. For the statistics of molecular groups, we assume that their concentrations are normally distributed, so the number of detected molecules is to some extent representative of the abundance of this molecular group. Because of the uncertainty introduced by sulfur-containing molecules, we revised and used statistical methods based on the number of molecules rather than molecular intensities when analyzing the proportions of different molecular groups. The related revisions are presented in Figures 4, S9, S10, S11, S12, and S17.

[Figure]

**Figure R3. (a)** Correlation analysis between the number proportion of carbonyl and non-carbonyl molecules and $DTT_{OC}$. **(b)** Correlation analysis between the number of carbonyl and non-carbonyl molecules and $DTT_{OC}$. **(c)** Correlation analysis between the normalized molecular intensities of carbonyl and non-carbonyl molecules (sulfur-containing molecules were eliminated) and $DTT_{OC}$.

**Reference:**

Xie, Q., Su, S., Dai, Y., Hu, W., Yue, S., Cao, D., Jiang, G., and Fu, P.: Deciphering 13C and 34S Isotopes of Organosulfates in Urban Aerosols by FT-ICR Mass Spectrometry, Environ. Sci. Technol. Lett., 9, 526–532, https://doi.org/10.1021/acs.estlett.2c00255, 2022.

7. *Lines 251-254: what molecular categories would the three sets of molecules described here from prior work be classified into if they were in Figure 1a? For me, the text here led to some dissonance with the categories used in Figure 1a; for example, I know 1,4-benzoquinone (a quinone) is an unsaturated hydrocarbon,*

*but based on O/C and H/C it would be considered aromatic. Similarly, molecules from Han et al. (2020) are described as "unsaturated," which is true, but these naphthalene-derivatives do not appear as though they would fall into the "unsaturated hydrocarbons" category of Figure 1. Please add additional context to the descriptions of molecules from prior work in relation to how chemical structures are categorized in this work.*

**Response:** Thanks for your suggestion. The previous seven molecular categories were replaced by five categories. In previous studies, the contributions of carbonyl compounds with different structures to the oxidative potential of OA by a variety of cellular and acellular assays have been revealed. Conjugated carbonyls such as benzaldehyde, α,β-unsaturated carbonyl, aromatic aldehydes, and polycyclic aromatic hydrocarbon o-quinones, mainly belong to polycyclic aromatic-like carbonyl molecules and highly aromatic-like carbonyl molecules according to van Krevelen diagrams, are found to have high oxidative potential and may cause the toxicities of OA. Revision at lines 278-281.

8. *Line 266: Might NaBH4 treatment induce any other reactions? If so, please note here, as well as whether these reactions likely have relevance to molecular structures in atmospheric OA.*

**Response:** Thanks for your comments. NaBH$_4$ is a widely used reagent to irreversibly reduce ketone, aldehyde, and quinone to alcohols. Previous studies have used various model compounds with other functional groups to react with NaBH$_4$, the results indicated that NaBH$_4$ can reduce carbonyls to alcohols with high selectivity (Baluha et al., 2013; Phillips and Smith, 2015; Ma et al., 2010). Revision at lines 292-293.

**Reference:**

Baluha, D. R., Blough, N. V., and Del Vecchio, R.: Selective Mass Labeling for Linking the Optical Properties of Chromophoric Dissolved Organic Matter to Structure and Composition via Ultrahigh Resolution Electrospray Ionization Mass Spectrometry, Environ. Sci. Technol., 47, 9891–9897, https://doi.org/10.1021/es402400j, 2013.

Phillips, S. M. and Smith, G. D.: Further Evidence for Charge Transfer Complexes in Brown Carbon Aerosols from Excitation–Emission Matrix Fluorescence Spectroscopy, J. Phys. Chem. A, 119, 4545–4551, https://doi.org/10.1021/jp510709e, 2015.

Ma, J., Del Vecchio, R., Golanoski, K. S., Boyle, E. S., and Blough, N. V.: Optical Properties of Humic Substances and CDOM: Effects of Borohydride Reduction, Environ. Sci. Technol., 44, 5395–5402, https://doi.org/10.1021/es100880q, 2010.

9. *Line 268: Were ambient-collected OA samples not available for this analysis? My recollection is that SRNOM has some similarities to OA from biomass burning and cloud-water samples but is not a perfect analogue.*

**Response:** Thanks for your great suggestion. Because of the limited amount of WSOM in $PM_{2.5}$ samples, it cannot meet the needs of $NaBH_4$ reduction, PFBHA derivatization, FT-ICR MS analysis, and DTT activity assay. Therefore, we employed two model samples for these analyses instead of real samples. One is SRNOM obtained from IHSS because it has similar chemical composition to HULIS (Havers et al., 1998; Graber and Rudich, 2006); The other is WSOM extracted from diesel soot, as it is an important source of carbonyls in urban OA (Grosjean et al., 2001; Wang et al., 2023a). We have revised and explained the reason for choosing these two models at lines 295-298.

In addition, we have performed additional experiment to verify whether $NaBH_4$ treatment can reduce the DTT activity using ambient-collected OA samples ($PM_{2.5}$ = 52.55 µg m$^{-3}$). As shown in Figure R4, after $NaBH_4$ treatment, the DTT activity of the real OA sample reduced by about 60%, which was consistent with the results using model samples. The corresponding discussion was added at lines 308-311 and Figure S15 of the revised manuscript.

[Figure]

**Figure R4. The DTT activity of WSOM from $PM_{2.5}$ sample before and after $NaBH_4$ treatment was measured.**

**Reference:**

Havers, N., Burba, P., Lambert, J., and Klockow, D.: Spectroscopic Characterization of Humic-Like Substances in Airborne Particulate Matter, Journal of Atmospheric Chemistry, 29, 45–54, https://doi.org/10.1023/A:1005875225800, 1998.

Graber, E. R. and Rudich, Y.: Atmospheric HULIS: How humic-like are they? A comprehensive and critical review, Atmos. Chem. Phys., 6, 729–753, https://doi.org/10.5194/acp-6-729-2006, 2006.

Grosjean, D., Grosjean, E., and Gertler, A. W.: On-Road Emissions of Carbonyls from Light-Duty and Heavy-Duty Vehicles, Environ. Sci. Technol., 35, 45–53, https://doi.org/10.1021/es001326a, 2001.

Wang, L., Wen, W., Yan, J., Zhang, R., Li, C., Jiang, H., Chen, S., Pardo, M., Zhu, K., Jia, B., Zhang, W., Bai, Z., Shi, L., Cheng, Y., Rudich, Y., Morawska, L., and Chen, J.: Influence of Polycyclic Aromatic Compounds and Oxidation States of Soot Organics on the Metabolome of Human-Lung Cells (A549): Implications for Vehicle Fuel Selection, Environ. Sci. Technol., 57, 21593–21604, https://doi.org/10.1021/acs.est.3c05228, 2023a.

*10. Line 275: would some amount of non-removal be expected in this type of analysis? Does this indicate anything about those unreacted molecular structures?*

**Response:** Thanks for your comments. NaBH$_4$ has been reported to selectively reduce the carbonyl groups present in aldehydes, ketones, and quinones to alcohols, but the reaction efficiencies between NaBH$_4$ with different carbonyls are different. SRNOM and WSOM extracted from diesel soot are mixtures containing thousands of compounds. The carbonyl compounds in these samples displayed high chemical diversity, resulting in a large difference in the efficiency of their reactions with NaBH$_4$, so some carbonyl molecules are not reduced under our experimental conditions. We did not further explore what structures of carbonyl molecules do not react with NaBH$_4$, because this is beyond the topic of this study. Nevertheless, this is a very interesting but very challenging topic that is worth exploring in depth in the future.

*11. Lines 320-321: the concluding sentence to this paragraph needs additional context. I would agree that the results here suggest aromatic-derived carbonyl molecules merit further attention, but there are a very low number of carbonyl-containing molecules in the "aromatic molecules" category on the VK diagram according to the data presented in Figures S8 and S9. However, the lignin-like and tannins-like categories are also defined around structures that contain aromatic rings, among other functionalities, and would likely have high AI and DBE values as well. Please clarify and update the text accordingly.*

**Response:** Thanks for your comments. We have updated the category definitions in the Van Krevelen diagram (see the 5th question). The "aromatic carbonyls" in the concluding sentence is actually intended to convey that "carbonyl molecules with

AImod > 0.5" in this study, which belong to the highly unsaturated-like, highly aromatic-like and polycyclic aromatic-like molecules in the Van Krevelen diagram, and that their share of all carbonyls is 60.8% ± 2.4%, 10.6% ± 4.9%, and 3.1% ± 1.2% (as shown in Figure R5 and R6).

We revised "aromatic molecules" to "highly unsaturated and aromatic carbonyls" at lines 358-359 of the revised manuscript.

[Figure]

**Figure R5. The proportion of molecules in five categories was estimated based on molecular numbers. Organic molecules detected in FT-ICP MS were distinguished into five compounds based on H/C, O/C, and AImod.**

[Figure]

**Figure R6. Five categories differ in the percentage distribution of all organic, non-carbonyl, and carbonyl molecules based on the molecular numbers (Kruskal-Wallis tests, Differences between groups were considered statistically significant when $p < 0.05$, with $0.01 < p < 0.05$ marked by *; $p < 0.01$ marked by **; and $p < 0.001$ marked by ***).**

*12. Lines 328-329: I'm not certain that the analysis presented in Figure 5b is entirely appropriate. The compound spaces defined in Kroll et al. (2011) and represented by the shaded spaces are derived from AMS measurements and volatility-based estimations of #C. As noted in Kroll et al., AMS measurements are imperfect due to the substantial fragmentation that occurs during analysis, leading to the loss of most detailed molecular information. Given the substantial differences in analytical techniques between this prior work and the current MS measurements that provide molecular formula with high accuracy, I'm not certain that a direct comparison in this manner is appropriate. Additionally, the O/C:#C space categories detailed in Kroll et al. are not exhaustive (as AMS measurements have advanced considerably in the years since) and do not include some aerosol types that may be relevant for this work, such as aged biomass OA, primary/fresh or secondary/aged OA from combustion related to residential heating, or cooking OA.*

**Response:** Thank you for your thoughtful comment. We acknowledge the limitations of the analysis presented in Figure 5b and the potential problems associated with directly comparing our MS measurements to the compound spaces defined by Kroll et al. In order to fully address the referee's concerns, we have deleted this figure.

*13. Line 339: can the authors be more specific regarding "combustion by-products?" Combustion can cover a variety of activities with different emissions profiles, including combustion related to vehicles, industry, power generation, or heating.*

**Response:** Thank you for your comment. Combustion by-products mainly consist of carbonyl-containing highly unsaturated and aromatic products from winter household heating activities (Steimer et al., 2020; Huang et al., 2018; Huo et al., 2021). Revised at lines 378-380 in the revised manuscript.

**Reference:**

Steimer, S. S., Patton, D. J., Vu, T. V., Panagi, M., Monks, P. S., Harrison, R. M., Fleming, Z. L., Shi, Z., and Kalberer, M.: Differences in the composition of organic aerosols between winter and summer in Beijing: a study by direct-infusion ultrahigh-resolution mass spectrometry, Atmospheric Chemistry and Physics, 20, 13303–13318, https://doi.org/10.5194/acp-20-13303-2020, 2020.

Huang, R.-J., Yang, L., Cao, J., Chen, Y., Chen, Q., Li, Y., Duan, J., Zhu, C., Dai, W., Wang, K., Lin, C., Ni, H., Corbin, J. C., Wu, Y., Zhang, R., Tie, X., Hoffmann, T., O'Dowd, C., and Dusek, U.: Brown Carbon Aerosol in Urban Xi'an, Northwest China: The Composition and Light Absorption Properties, Environ. Sci. Technol., 52, 6825–6833, https://doi.org/10.1021/acs.est.8b02386, 2018.

Huo, Y., Guo, Z., Li, Q., Wu, D., Ding, X., liu, A., Huang, D., Qiu, G., Wu, M., Zhao, Z., Sun, H., Song, W., Li, X., Chen, Y., Wu, T., and Chen, J.: Chemical Fingerprinting of HULIS in Particulate Matters Emitted from Residential Coal and Biomass Combustion, Environ. Sci. Technol., 55, 3593–3603, https://doi.org/10.1021/acs.est.0c08518, 2021.

***Section 4***

*14. Lines 356-368: The discussion in this paragraph feels odd, given that the first sentence highlights the importance of aromatic carbonyl molecules for OP and the rest of the paragraph focuses on the reactions of non-aromatic carbonyls that, as best I can tell, do not contribute strongly to OP; consider revising. The oligomerization reactions highlighted here tend to produce molecules with high O:C ratio and a range of H:C ratios, depending on the reactions, that would appear to place these products mostly in the tannins and carbohydrates categories on the VK diagram. The authors observe plenty of carbonyls in these categories (Figure 1a) but almost none that are strongly associated with OP (Figure 5a), which I suggest the authors also discuss.*

**Response:** Thank you for your insightful comments. The conclusion of the paper was rewritten to highlight our main research results and findings. Specifically, we highlight the significant contribution of highly unsaturated and aromatic carbonyl molecules to OP, indicating the need for more attention to the potential health risks posed by these molecules. These carbonyls primarily originate from fuel combustion, including biomass burning and fossil fuel combustion. The detailed revision was presented at lines 393-397 of the revised manuscript.

According to the updated category definitions in the Van Krevelen diagram, carbonyl molecules formed through oligomerization reactions with high O/C ratios and a range of H/C ratios are categorized as unsaturated aliphatic-like ($1.5 \leqslant$ H/C $<$ 2.0) and saturated-like molecules (H/C $\geqslant$ 2.0 or O/C $\geqslant$ 0.8), comprising 25.1% $\pm$ 6.1% and 0.4% $\pm$ 0.1% of all carbonyls, significantly less than highly unsaturated and aromatic carbonyls. As discussed in Section 3.5 at lines 353-355, carbonyl compounds with different structures should have different DTT activities. O/C$_w$, and H/C$_w$ were

significantly negatively correlated with $DTT_{OC}$ (Figure S19). Therefore, the contribution of unsaturated aliphatic-like and saturated-like molecules to oxidative potential is minimal. The corresponding discussion was added at lines 356-358.

15. *Lines 372-373: is the statement that "no significant decrease in atmospheric carbonyls during the Winter Olympics was observed" based on the number of unique carbonyl molecules observed or the absolute signal apportioned to carbonyl molecules? Based on the large drop in OA mass between the winter and Winter Olympics periods I would be surprised if the amount of carbonyl molecules within OA remained constant.*

**Response:** Thank you for your comments. Our statement is based on the observed number of carbonyl molecules. The number of carbonyl molecules decreased slightly during the Winter Olympics (average $_{number}$ = 1643) compared to winter (average $_{number}$ = 1669). Furthermore, the normalized molecular intensities of carbonyl molecules follow the same trend, with a slight decrease during the Winter Olympics (average = 27.8%) compared to winter (average = 28.3%). We clarified this statement at lines 397-398 of the manuscript.

16. *Lines 373-375: I agree with this statement, but it also seems to me that, based on my understanding of the data, the authors can give more detail on likely sources of carbonyl-derived OP in the collected aerosol samples. The authors demonstrate a link between OP and carbonyl molecules with aromatic content (based on AI and DBE) and show a lack of decrease in OP following emissions reduction during the Winter Olympics when traditional OA sources such as industry and transport decreased but other sources like heating (or potentially cooking?) did not. To me, this suggests that combustion related to residential heating may be a significant source of carbonyl molecules that contribute to OP, as the authors mention at Line 240, and that changes to residential heating techniques or air quality may reduce the OP of OA. Are there previous studies that examine changes in air quality and OA composition during the Winter Olympics period that might inform the authors' discussion on OP during the winter and Winter Olympic periods?*

**Response:** Thank you for your constructive suggestions. Benefitting from prior investigations into the characteristics and sources of $PM_{2.5}$ during the Winter

Olympics (Liu et al., 2022), we have learned that transportation and industrial emissions decreased during the Winter Olympic. However, the Winter Olympics did not impose restrictions on residential heating and widespread cooking activities. Therefore, livelihood-related heating and household emissions contribute significantly to the abundance of atmospheric carbonyl molecules, leading to elevated OP in atmospheric aerosols under the current air quality level in Beijing. Heating and cooking may serve as important sources of OP or carbonyl molecules during the Winter Olympics.

In particular, this study observed a significantly lower $DTT_v$ level in 2022 compared to those reported in 2016 (1.9±0.3 nmol min$^{-1}$ m$^{-3}$) and 2017 (5.8±7.1 nmol min$^{-1}$ m$^{-3}$) for Beijing $PM_{2.5}$ samples (Yu et al., 2022; Campbell et al., 2021). This may be attributed to China's implementation of the "Three-Year Action Plan (2018-2020)", an energy policy that gradually shifted the use of coal to natural gas and electricity for cooking and heating in rural areas, consequently reducing household emissions (Du et al., 2022; Li et al., 2023c). These findings underscore the importance of addressing residential heating and cooking practices to mitigate OP levels in atmospheric aerosols during the winter period. We added the discussion in the revised manuscript at lines 247-257 and 398-400.

**Reference:**

Liu, Y., Xu, X., Yang, X., He, J., Ji, D., and Wang, Y.: Significant Reduction in Fine Particulate Matter in Beijing during 2022 Beijing Winter Olympics, Environ. Sci. Technol. Lett., https://doi.org/10.1021/acs.estlett.2c00532, 2022.

Yu, Q., Chen, J., Qin, W., Ahmad, M., Zhang, Y., Sun, Y., Xin, K., and Ai, J.: Oxidative potential associated with water-soluble components of PM2.5 in Beijing: The important role of anthropogenic organic aerosols, J. Hazard. Mater., 433, 128839, https://doi.org/10.1016/j.jhazmat.2022.128839, 2022.

Campbell, S. J., Wolfer, K., Utinger, B., Westwood, J., Zhang, Z.-H., Bukowiecki, N., Steimer, S. S., Vu, T. V., Xu, J., Straw, N., Thomson, S., Elzein, A., Sun, Y., Liu, D., Li, L., Fu, P., Lewis, A. C., Harrison, R. M., Bloss, W. J., Loh, M., Miller, M. R., Shi, Z., and Kalberer, M.: Atmospheric conditions and composition that influence PM2.5 oxidative potential in Beijing, China, Atmospheric Chemistry and Physics, 21, 5549–5573, https://doi.org/10.5194/acp-21-5549-2021, 2021.

Du, H., Li, J., Wang, Z., Chen, X., Yang, W., Sun, Y., Xin, J., Pan, X., Wang, W., Ye, Q., and Dao, X.: Assessment of the effect of meteorological and emission variations on winter PM2.5 over the North China Plain in the three-year action plan against air pollution in 2018–2020, Atmospheric Research, 280, 106395, https://doi.org/10.1016/j.atmosres.2022.106395, 2022.

Li, Y., Lei, L., Sun, J., Gao, Y., Wang, P., Wang, S., Zhang, Z., Du, A., Li, Z., Wang, Z., Kim, J. Y., Kim, H., Zhang, H., and Sun, Y.: Significant Reductions in Secondary Aerosols after the Three-Year Action Plan in Beijing Summer, Environ. Sci. Technol., 57, 15945–15955, https://doi.org/10.1021/acs.est.3c02417, 2023c.

17. *Technical comments: I suggest removing instances of informal language in the Introduction (e.g., "more and more," "tip of the iceberg,"). I also suggest adding a more complete description of the word "montane," as this word is not commonly used and may be unfamiliar to some readers.*

**Response:** Accepted, and all of these words were revised.

---

## Author Comment (AC2)

**Point-by-Point Responses to the Comments from Reviewers**

Manuscript ID: egusphere-2024-37

Manuscript Title: "Critical contribution of chemically diverse carbonyl molecules to the oxidative potential of atmospheric aerosols"

The corresponding authors: Prof. Jitao Lv and Prof. Yawei Wang

**Responses to the Comments from Anonymous Referee 2:**

***Comments:***

*Li et al., utilized FT-ICRMS technology with their established screening procedure enabling a high-throughput screening of carbonyl molecules in ambient aerosol samples. They further linked these data with the DTT activity of water-soluble organic matter and found a positive correlation between carbonyl molecules and DTT activity. By employing several statistical tests and modeling, they proposed oxidized aromatic compounds containing the carbonyl group could be used as potential markers of atmospheric oxidative stress. Overall, I think these data are beneficial and the results are of great importance. I would recommend publication after the following concerns are addressed.*

**Response:** We sincerely appreciate your positive comments. We have made careful revisions according to your comments and provided a point-by-point explanation for each comment.

1. *A significant correlation between $DTT_{OC}$ and the number of carbonyl molecules was observed; however, the molecular intensity of carbonyls was not taken into account. I suggest the authors should consider weighing the effects of intensity and examine how it may affect the correlation.*

**Response:** Thanks for your suggestion, your point is similar to that of reviewer 1's comment 6.

Both the number and the number proportion of carbonyl molecules were positively correlated with $DTT_{OC}$ (Pearson's r = 0.55, $p < 0.01$; and Pearson's r = 0.63, $p < 0.001$), whereas an insignificant or negative correlation was observed between those of non-carbonyls and $DTT_{OC}$ (Pearson's r = -0.26, $p = 0.185$; and Pearson's r = -0.63, $p < 0.001$). We also performed a correlation analysis between $DTT_{OC}$ and the normalized molecular intensities of carbonyls or non-carbonyls of each sample. Insignificant correlation was observed between the normalized molecular intensities of all carbonyls or non-carbonyls and $DTT_{OC}$. However, a significant positive correlation between the normalized molecular intensities of carbonyls and $DTT_{OC}$ was obtained (Pearson's r = 0.56, $p < 0.01$) when sulfur-containing molecules were excluded (Figure R1 c). Sulfur-containing molecules in aerosols often displayed extremely high ionization efficiencies, leading to unreasonably high intensities of S-containing molecules detected by FT-ICR MS (Xie et al., 2022). When considering normalized molecular intensities in correlation analysis, it is more reasonable to exclude sulfur-containing molecules. The related revision and discussion have been added at lines 268-271 and Figure S14 in the revised manuscript.

In fact, it is a great challenge to quantification of mass data obtained by FT-ICR MS, as the peak intensity in a mass spectrum is not only determined by the initial concentration but also by the ionization efficiency of molecules. Using statistics based on the number of molecules can avoid the effect of ionization efficiency, but the concentration difference between molecules is ignored. For the statistics of molecular groups, we assume that their concentrations are normally distributed, so the number of detected molecules is to some extent representative of the abundance of this molecular group. Because of the uncertainty introduced by sulfur-containing molecules, we revised and used statistical methods based on the number of molecules rather than molecular intensities when analyzing the proportions of different molecular groups. The related revisions are presented in Figures 4, S9, S10, S11, S12, and S17.

[Figure]

**Figure R1.** (a) Correlation analysis between the number proportion of carbonyl and non-carbonyl molecules and $DTT_{OC}$. (b) Correlation analysis between the number of carbonyl and non-carbonyl molecules and $DTT_{OC}$. (c) Correlation analysis between the normalized molecular intensities of carbonyl and non-carbonyl molecules (sulfur-containing molecules were eliminated) and $DTT_{OC}$.

**Reference:**

Xie, Q., Su, S., Dai, Y., Hu, W., Yue, S., Cao, D., Jiang, G., and Fu, P.: Deciphering 13C and 34S Isotopes of Organosulfates in Urban Aerosols by FT-ICR Mass Spectrometry, Environ. Sci. Technol. Lett., 9, 526–532, https://doi.org/10.1021/acs.estlett.2c00255, 2022.

2. *The authors utilized Suwannee River natural organic matter and diesel soot to validate a decrease in DTT activity after the removal of carbonyls. I suggest the authors should discuss why similar experiments were not performed using real*

*ambient WSOM. If feasible, conducting additional experiments to demonstrate a substantial decrease in DTT activity after removing carbonyls for ambient WSOM is recommended. Such results would enhance the credibility of the findings. At the very least, the representatives of Suwannee River natural organic matter and diesel soot should be discussed in the manuscript.*

**Response:** Thanks for your thoughtful comments. Similar to comment 9 of reviewer 1, we have performed additional experiment to verify whether $NaBH_4$ treatment can reduce the DTT activity using ambient-collected OA samples ($PM_{2.5}$ = 52.55 µg m$^{-3}$). As shown in Figure R2, after $NaBH_4$ treatment, the DTT activity of the real OA sample reduced by about 60%, which was consistent with the results using model samples. And the corresponding discussion was added at lines 308-311 and Figure S15 of the revised manuscript.

Because of the limited amount of WSOM in $PM_{2.5}$ samples, it cannot meet the needs of $NaBH_4$ reduction, PFBHA derivatization, FT-ICR MS analysis, and DTT activity assay. Therefore, we employed two model samples for these analyses instead of real samples. One is SRNOM obtained from IHSS because it has similar chemical composition to HULIS (Havers et al., 1998; Graber and Rudich, 2006); The other is WSOM extracted from diesel soot, as it is an important source of carbonyls in urban OA (Grosjean et al., 2001; Wang et al., 2023a). We have revised and explained the reason for choosing these two models at lines 295-298.

[Figure]

**Figure R2. The DTT activity of WSOM from $PM_{2.5}$ sample before and after NaBH4 treatment was measured.**

**Reference:**

Havers, N., Burba, P., Lambert, J., and Klockow, D.: Spectroscopic Characterization of Humic-Like Substances in Airborne Particulate Matter, Journal of Atmospheric Chemistry, 29, 45–54, https://doi.org/10.1023/A:1005875225800, 1998.

Graber, E. R. and Rudich, Y.: Atmospheric HULIS: How humic-like are they? A comprehensive and critical review, Atmos. Chem. Phys., 6, 729–753, https://doi.org/10.5194/acp-6-729-2006, 2006.

Grosjean, D., Grosjean, E., and Gertler, A. W.: On-Road Emissions of Carbonyls from Light-Duty and Heavy-Duty Vehicles, Environ. Sci. Technol., 35, 45–53, https://doi.org/10.1021/es001326a, 2001.

Wang, L., Wen, W., Yan, J., Zhang, R., Li, C., Jiang, H., Chen, S., Pardo, M., Zhu, K., Jia, B., Zhang, W., Bai, Z., Shi, L., Cheng, Y., Rudich, Y., Morawska, L., and Chen, J.: Influence of Polycyclic Aromatic Compounds and Oxidation States of Soot Organics on the Metabolome of Human-Lung Cells (A549): Implications for Vehicle Fuel Selection, Environ. Sci. Technol., 57, 21593–21604, https://doi.org/10.1021/acs.est.3c05228, 2023a.

*3. The reported DTTm (Figure 2) is over an order of magnitude lower than the general reported DTTm in literature (e.g., Yu et al., Journal of Hazardous Materials, 2022 (10.1016/j.jhazmat.2022.128839); Campbell et al., Atmos. Chem. Phys., 2021 (10.5194/acp-21-5549-2021), works for Beijing DTT activities). Please discuss the possible reasons for the difference.*

**Response:** Thanks for your suggestion. After recalculating for $DTT_m$ (see the 4th question), the $DTT_m$ range in our study is between 0.2 and 28.2 pmol/min/µg, which is consistent with the order of magnitude reported in previous literature. Figure 2 and S13 have depicted the correct values of $DTT_m$.

In addition, the DTT levels measured in this study were still reduced compared to previous studies. For example, this study observed a significantly lower DTT level in 2022 compared to those reported in 2016 ($DTT_v$ = 1.9±0.3 nmol min$^{-1}$ m$^{-3}$, $DTT_m$ ≈ 10-75 pmol min$^{-1}$ µg$^{-1}$) and 2017 (5.8±7.1 nmol min$^{-1}$ m$^{-3}$, $DTT_m$ ≈ 20-160 pmol min$^{-1}$ µg$^{-1}$) for Beijing PM2.5 samples (Yu et al., 2022; Campbell et al., 2021). This may be attributed to China's implementation of the "Three-Year Action Plan (2018-2020)", an energy policy that gradually shifted the use of coal to natural gas and electricity for cooking and heating in rural areas, leading to livelihood-related heating and household emissions were reduced (Du et al., 2022; Li et al., 2023c). The revision was presented at lines 251-257 of the revised manuscript.

**Reference:**

Yu, Q., Chen, J., Qin, W., Ahmad, M., Zhang, Y., Sun, Y., Xin, K., and Ai, J.: Oxidative potential associated with water-soluble components of PM2.5 in Beijing: The important role of anthropogenic organic aerosols, J. Hazard. Mater., 433, 128839, https://doi.org/10.1016/j.jhazmat.2022.128839, 2022.

Campbell, S. J., Wolfer, K., Utinger, B., Westwood, J., Zhang, Z.-H., Bukowiecki, N., Steimer, S. S., Vu, T. V., Xu, J., Straw, N., Thomson, S., Elzein, A., Sun, Y., Liu, D., Li, L., Fu, P., Lewis, A. C., Harrison, R. M., Bloss, W. J., Loh, M., Miller, M. R., Shi, Z., and Kalberer, M.: Atmospheric conditions and composition that influence PM2.5 oxidative potential in Beijing, China, Atmospheric Chemistry and Physics, 21, 5549–5573, https://doi.org/10.5194/acp-21-5549-2021, 2021.

Du, H., Li, J., Wang, Z., Chen, X., Yang, W., Sun, Y., Xin, J., Pan, X., Wang, W., Ye, Q., and Dao, X.: Assessment of the effect of meteorological and emission variations on winter PM2.5 over the North China Plain in the three-year action plan against air pollution in 2018–2020, Atmospheric Research, 280, 106395, https://doi.org/10.1016/j.atmosres.2022.106395, 2022.

Li, Y., Lei, L., Sun, J., Gao, Y., Wang, P., Wang, S., Zhang, Z., Du, A., Li, Z., Wang, Z., Kim, J. Y., Kim, H., Zhang, H., and Sun, Y.: Significant Reductions in Secondary Aerosols after the Three-Year Action Plan in Beijing Summer, Environ. Sci. Technol., 57, 15945–15955, https://doi.org/10.1021/acs.est.3c02417, 2023c.

4. *Furthermore, based on the discrepancy between DTTv and DTTm in Figure 2, it appears that they cannot be converted by the equation provided by the author. For instance, to obtain the DTTv shown in Figure 2, the DTTm would need to be multiplied by approximately 1000 ug/m3 PM2.5 concentration. I recommend double-checking the data for accuracy. Additionally, please include the PM2.5 concentrations in Table S1, and it would be preferable to provide TOC concentration data for the air rather than in the WSOM extracts.*

**Response:** Thanks for your insightful comments. We have provided $PM_{2.5}$ concentration data ($\mu g\ m^{-3}$) at Table S1 and Figure S8.

We meticulously reviewed the calculation process of converting $DTT_v$ to $DTT_m$, and we found a mistake when during conversion. In the previous manuscript, we used the product of $PM_{2.5}$ concentration and 24-hour sampling time for calculating $DTT_m$. In the revised manuscript, only the $PM_{2.5}$ concentration was used for $DTT_m$ calculation, consistent with the equation provided at line 171. Additionally, $DTT_{OC}$ utilized the TOC concentration of WSOC, thus remaining unaffected.

In the revised manuscript, Figures 2 and S13 depicted the correct values of $DTT_m$ (0.2-28.2 pmol $min^{-1}$ $\mu g^{-1}$). Additionally, we only measured the TOC values of WSOM, hence cannot provide TOC concentration data for the air data.

5. *Figure S11 will be more informative by showing the categorized groups of molecular formulas for different sampling periods instead.*

**Response:** Thanks for your suggestion. The modified figure illustrates a comparison of the categorization of different molecular formulas in the three sampling periods (Winter, Winter Olympics, and Summer) (shown in Figure R3 below). The corresponding modifications are shown in Figure S12 in the Supporting Information.

[Figure]

**Figure R3. Differences in the percentage distribution of elemental composition based on the molecular numbers of all organic, non-carbonyl, and carbonyl molecules in winter, summer and Winter Olympics (Kruskal-Wallis tests, Differences between groups were considered statistically significant when $p < 0.05$, with $0.01 < p < 0.05$ marked by *; $p < 0.01$ marked by **; and $p < 0.001$**

marked by ***).

6. *Line 244: "... This is likely due to the differences in aerosol sources in different seasons." please expand the discussion, e.g., what could be the main sources for DTT in summer and winter, respectively.*

**Response:** Thank you for your suggestion. In winter, fossil fuel and biomass burning is a major local source of PM$_{2.5}$ in suburban sites, while this special combustion source aerosol in suburban sites was eliminated in summer (Steimer et al., 2020; Campbell et al., 2021). The relevant discussion was added at lines 261-263 of the revised manuscript.

**Reference:**

Steimer, S. S., Patton, D. J., Vu, T. V., Panagi, M., Monks, P. S., Harrison, R. M., Fleming, Z. L., Shi, Z., and Kalberer, M.: Differences in the composition of organic aerosols between winter and summer in Beijing: a study by direct-infusion ultrahigh-resolution mass spectrometry, Atmospheric Chemistry and Physics, 20, 13303–13318, https://doi.org/10.5194/acp-20-13303-2020, 2020.

Campbell, S. J., Wolfer, K., Utinger, B., Westwood, J., Zhang, Z.-H., Bukowiecki, N., Steimer, S. S., Vu, T. V., Xu, J., Straw, N., Thomson, S., Elzein, A., Sun, Y., Liu, D., Li, L., Fu, P., Lewis, A. C., Harrison, R. M., Bloss, W. J., Loh, M., Miller, M. R., Shi, Z., and Kalberer, M.: Atmospheric conditions and composition that influence PM2.5 oxidative potential in Beijing, China, Atmospheric Chemistry and Physics, 21, 5549–5573, https://doi.org/10.5194/acp-21-5549-2021, 2021.

7. *Line 338: "These results suggested that aromatic secondary products containing carbonyl group produced from combustion by-products in winter are potential molecular markers of atmospheric oxidative stress." Here I suggest the authors discuss the study by Liu et al., es&t, 2023 (10.1021/acs.est.3c03641), where similar findings are observed for cellular oxidative stress. Also, the authors should clarify the combustion activities only occurred in winter (and also during the winter Olympic period) but not in summer.*

Response: Thank you for your comment and provide us with an excellent reference. According to Liu et al., we further highlight the role of oxygenated organic aerosols containing carbon-oxygen double bonds and aromatic structures in cellular oxidative stress, along with their higher contribution during the winter season. The revision was presented at lines 376-378 of the revised manuscript.

Additionally, while combustion activities are not exclusively limited to the winter season, they typically increase during winter due to heating activities in cold conditions. We clarified this point in lines 379-380 of the revised manuscript.

*8. Conclusion section: the first paragraph regarding the authors' FT-ICR-MS method was not discussed in the Results and Discussion section and is not rational to be a conclusion of this work.*

**Response:** Thank you for your suggestion. The section regarding the FT-ICR-MS method has been deleted.

*9. Conclusion section: aromatic carbonyl molecules are suggested as indicators of atmospheric oxidative stress. However, the second paragraph primarily discusses the possible formation sources of general carbonyl molecules, such as terpene and isoprene oxidation, which do not produce aromatic carbonyl molecules. The authors should revise the discussion to be more explicit and focused. Otherwise, the current discussion on the formation pathways of carbonyls seems to favor summer conditions, which contradicts the findings of this study.*

**Response:** Thank you for your insightful suggestion. The conclusion of the paper was rewritten to highlight our main research results and findings. Specifically, we highlight the significant contribution of highly unsaturated and aromatic carbonyl molecules to OP, indicating the need for more attention to the potential health risks posed by these molecules. These carbonyls primarily originate from fuel combustion, including biomass burning and fossil fuel combustion. The detailed revision was presented at lines 393-397of the revised manuscript.

*Minor comments:*

1. *In Figure S2 (a) and (b), the fittings for blank samples look incorrect. The fitting curves do not come across any data point.*

**Response:** We performed linear regression on the original blank sample data once again in the origin software, and obtained a new fitted curve (Figure R4), as shown in Figure S3 in the revised supporting information.

[Figure]

**Figure R4. Reproducibility of DTT measurements of samples. (a) DTT measurements of PM$_{2.5}$ samples and backgrounds, using the downtown site on January 27, 2022, as an example. (b) DTT measurements of a water-soluble organic carbon (WSOM) quality control sample, which was prepared containing a mixture of all WSOM samples. (c) DTT measurements of a water-insoluble organic carbon (WISOM) quality control sample, which was prepared containing a mixture of all WISOM samples.**

2. *Please clarify the sampling period for each sample. Was it 24-hr sampling?*

**Response:** The sampling period for each sample was one day (24 hours), which was added to line 101 of the revised manuscript.

3. *A number of typos in the manuscript, please carefully check. E.g., WSOM has been written as WOSM here and there.*

**Response:** The manuscript has been carefully examined and all "WOSM" has been revised to "WSOM", as at lines 133, 139 and 290.

4. *Figure S11, "CHO" figure, x-axis, "oC=O" should be ""no C=O*

**Response:** The x-axis label "oC=O" has been corrected to "no C=O" in Figure S12.